# Explicit modelling of isoprene chemical processing in polluted air masses in suburban areas of the Yangtze River Delta region: radical cycling and formation of ozone and formaldehyde

**Kun Zhang [a, b], Ling Huang [a, b], Qing Li [a, b], Juntao Huo [c], Yusen Duan [c], Yuhang Wang [d], Elly Yaluk [a, b], Yangjun Wang [a, b], Qingyan Fu [c], Li Li [a, b*]**

[a] School of Environmental and Chemical Engineering, Shanghai University, Shanghai, 200444, China

[b] Key Laboratory of Organic Compound Pollution Control Engineering, Shanghai University, Shanghai, 200444, China

[c] Shanghai Environmental Monitoring Center, Shanghai, 200235, China

[d] School of Earth and Atmospheric Sciences, Georgia Institute of Technology, Atlanta, GA, USA

*Correspondence to* Li Li (Lily@shu.edu.cn)

## Abstract

In recent years, ozone pollution has become among the most severe environmental problems in China. Evidence from observations have showed increased frequency of high $O_3$ levels in suburban areas of the Yangtze River Delta (YRD) region. To better understand the formation mechanism of local $O_3$ pollution and investigate the potential role of isoprene chemistry in the budgets of RO$x$ (OH+HO$_2$+RO$_2$) radicals, synchronous observations of volatile organic compounds (VOCs), formaldehyde (HCHO) and meteorological parameters were conducted at a suburban site of the YRD region in 2018. Five episodes with elevated $O_3$ concentrations under stagnant meteorological conditions were identified; an observation-based model (OBM) with the Master Chemical Mechanism was applied to analyze the photochemical processes during these high $O_3$ episodes. The high levels of $O_3$, nitrogen oxides (NO$x$), and VOCs facilitated strong production and recycling of RO$x$ radicals with the photolysis of

oxygenated VOCs (OVOCs) being the primary source. Our results suggest that, local biogenic
isoprene is important in suburban photochemical processes. Removing isoprene could
drastically slow down the efficiency of RO*x* recycling and reduce the concentrations of RO*x*.
Besides, the absence of isoprene chemistry could further lead to decrease in the daily average
concentration of $O_3$ and HCHO by 34% and 36%, respectively. Therefore, this study
emphasizes the importance of isoprene chemistry in suburban atmosphere, particularly with
the participation of anthropogenic NO*x*. Moreover, our results provide insights into the radical
chemistry that essentially drives the formation of secondary pollutants (e.g. $O_3$ and HCHO) in
the suburban of YRD region.
**Keywords:** Isoprene; Observation-based model (OBM); Radical; Ozone; Yangtze River Delta

## 1. Introduction

The hydroxyl radical (OH), hydro peroxy radical ($HO_2$) and organic peroxy radical ($RO_2$),
collectively known as RO*x* dominate the oxidative capacity of the atmosphere and hence
govern the removal of primary contaminants (e.g. volatile organic compounds (VOCs)) and
the formation of secondary pollutants (e.g. ozone ($O_3$) , secondary organic aerosols (SOAs))
(Liu et al., 2012; Xue et al., 2016). RO*x* radicals can undergo efficient recycling (e.g. OH→
$RO_2$→RO→$HO_2$→OH) and produce $O_3$ and oxygenated VOCs (OVOCs) (Liu et al., 2012;
Tan et al., 2019; Xue et al., 2016). In addition, the photolysis of OVOCs can in turn produce
primary $RO_2$ and $HO_2$ radicals, and further accelerate the recycling of RO*x* (Liu et al., 2012).
The reaction rates of different VOCs with RO*x* vary significantly (Atkinson and Arey, 2003;
Atkinson et al., 2006). For instance, the reaction rate constants for OH with ethane and ethene
are $0.248 \times 10^{-12}$ (cm molecule$^{-1}$ s$^{-1}$) and $8.52 \times 10^{-12}$ (cm molecule$^{-1}$ s$^{-1}$), respectively. Among
the hundreds and thousands of VOC species, isoprene ($C_5H_8$, 2-methyl-1,3-butadiene) is
among the most active and abundant biogenic VOCs (BVOCs) species globally (Wennberg et

al., 2018). Over the past decades, isoprene emission sources have been extensively studied (Gong et al., 2018) and recent works have focused on the degradation pathways and the impact of isoprene chemistry on regional forest chemistry (Gong et al., 2018; Wolfe et al., 2016a). Previous studies showed that isoprene could be quickly oxidized by atmospheric oxidants (e.g. OH, $O_3$ or $NO_3$) (Wolfe et al., 2016a; Gong et al., 2018; Jenkin et al., 2015). Due to the rapid reaction between OH and isoprene ($100 \times 10^{-12}$ $cm^3$ $molecule^{-1}$ $s^{-1}$ at 298 K), more than 90% of the total daytime isoprene is removed via this reaction (Wennberg et al., 2018). The reaction between OH and isoprene is initiated by the addition of OH and can generate isoprene hydroxyperoxy radicals ($ISOPO_2$) (Wennberg et al., 2018; D'Ambro et al., 2017; Liu et al., 2013; Jenkin et al., 2015). $ISOPO_2$ isomers could then interconvert rapidly due to reversible $O_2$ addition and are finally removed via reactions with $HO_2$ or NO (Jenkin et al., 2015; Wolfe et al., 2016a). Hence, the degradation process of isoprene is tightly associated with RO$x$ recycling. According to He et al. (2019), isoprene chemistry could strongly influence the photochemical formation of $O_3$, with a relative incremental reactivity (RIR) of ~0.06%/%. Besides, HCHO is also formed via several pathways during the depletion of isoprene (Jenkin et al., 2015; Wolfe et al., 2016a) and is found to be highly sensitive to isoprene emissions (Zeng et al., 2019).

The Yangtze River Delta (YRD) region is one of the most developed city-clusters in eastern China and is under serious $O_3$ pollution (Zhang et al., 2019; Zhang et al., 2020a; Chan et al., 2017). At the suburban area of YRD, high levels of $O_3$ have been frequently observed (Zhang et al., 2019; Zhang et al., 2020a). Several studies have investigated the relationship between $O_3$ and its precursors (Chan et al., 2017; Lin et al., 2020; Zhang et al., 2020a; Zhang et al., 2020b), but few studies have addressed the atmospheric oxidizing capacity and radical chemistry involved in these complicated photochemical processes (Tan et al., 2019; Zhu et al., 2020b). Previous studies have pointed out that the high levels of $O_3$ at suburban areas of

Shanghai could be attributed to the transport of $O_3$ or its precursors from urban areas (Lin et
al., 2020; Zhang et al., 2019; Li et al., 2016; Li et al., 2019). On the contrary, high $O_3$
concentrations were frequently observed in suburban areas under stable meteorological
conditions. Therefore, given the dense vegetation cover in suburban YRD and weak transport
of air masses, the importance of local isoprene chemistry regarding ozone formation remains
unclear.
In this study, we conducted a comprehensive set of in-situ observations of isoprene
concentration, meteorological conditions, and concentrations of atmospheric pollutants
(including $O_3$, NO$x$, CO, VOCs, and HCHO) to understand the impact of isoprene chemistry
on atmospheric photochemical processes in suburban YRD region. We used an observation-
based model (OBM) to explore the role of local isoprene chemistry in radical budgets and the
formation of $O_3$ and HCHO. Results from this study provides insights into the isoprene
chemistry in the suburban region of a fast-developing city-cluster.
## 2. Methodology
### 2.1 Field measurement
The observations were conducted at a supersite (120.98°E, 31.09°N) in the suburban areas
of the YRD region (Figure 1). It is located in the west of Shanghai and is close to the Dianshan
Lake Scenic area, with relatively higher vegetation cover than the urban areas. To investigate
the local isoprene chemistry and its influence on $O_3$ and HCHO formation, continuous
measurements were conducted from April 7[th] to September 25[th], 2018, when photochemical
activity and $O_3$ formation is significant.


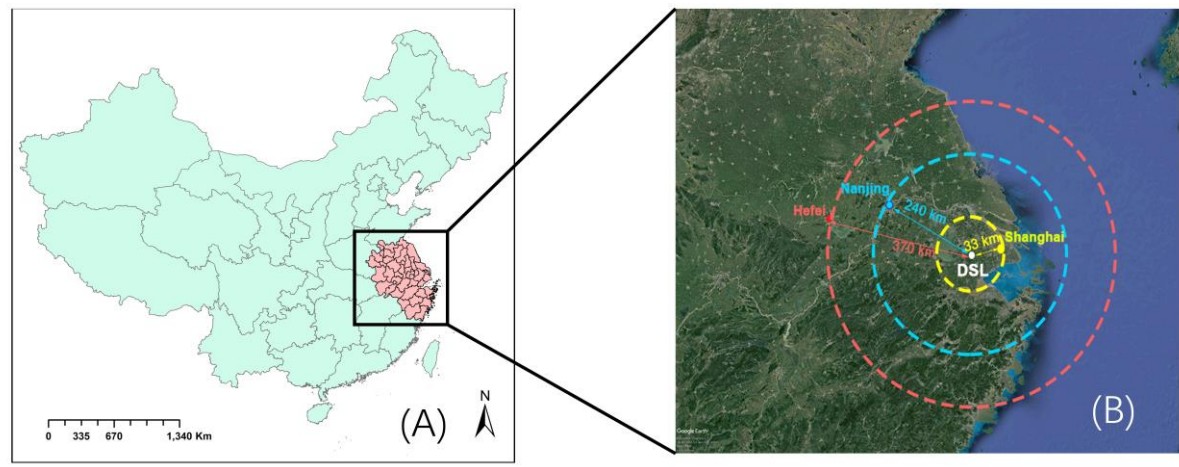


**Figure 1. (A) map of China with YRD region highlighted in pink; and (B) satellite map of YRD region**
**(created with Google Earth© on 23rd July 2020).**
**Table 1. Measurements performed during the ozone season.**

| Species/Parameter | Experimental Technique | Time resolution | Lower Detectable limit |
|---|---|---|---|
| $O_3$ | Model 49i, Thermo Fischer Scientific, USA | 60 s | 0.5 ppbv |
| NO and $NO_2$ | Model 42i, Thermo Fischer Scientific, USA | 60 s | 0.4 ppbv |
| CO | Model 48i, Thermo Fischer Scientific, USA | 60 s | 40 ppbv |
| HCHO | AL4021, Aero-Laser, GER | 90 s | 0.1 ppbv |
| VOCs species | GC866, Agilent., USA | 1 hour | - |
| Temperature, relative humidity, wind speed and wind direction | Meteorological station, Vaisala, FIN | 60 s | - |


The measuring instruments are shown in Table 1. Wind speed (WS), wind direction (WD),
ambient pressure (P), temperature (T), and relative humidity (RH) were simultaneously
observed by a meteorological station (Vaisala., FIN). According to China's air quality standard,
several criteria air pollutants were measured during this experiment. For instance, $O_3$ was
measured by an ultraviolet photometric analyzer (Model 49i, Thermo Fischer Scientific., USA),
with a detection limit of 0.5ppbv, whereas nitrogen oxides (NO and $NO_2$) were simultaneously
observed by a chemiluminescence instrument (Model 42i, Thermo Fischer Scientific., USA),
with a detection limit of 0.4ppbv. Likewise, carbon monoxide (CO) was monitored by a gas
filter correlation infrared absorption analyzer (Model 48i, Thermo Fischer Scientific., USA),
with a detection limit of 0.04ppm. All the online instruments used for gas analyzer were auto-
zero every day, and were multi-point calibrated every month. All the instruments used for the
online observation were housed on top of a 5-floor-high building, which was about 15 m above
the ground level.
A total of 55 VOC species, including 28 alkanes, 10 alkenes (including isoprene), 16
aromatics and acetylene were continuously analyzed at our sampling site by two online gas
chromatographs with flame ionization detector (GC-FID) systems (GC-866 airmoVOC $C_2$-$C_6$
#58850712 and airmoVOC $C_6$-$C_{12}$ #283607112, Agilent., USA) with a time resolution of 1
hour during the study period. Ambient samples are directly inhaled into this system by a pump.
Low carbon VOCs ($C_2$-$C_6$) are captured by a low temperature (-10 °C) pre-concentration
system, while high carbon VOCs are concentrated by a built-in room temperature pre-
concentration system. Then the preconcentration system are heated and desorb VOCs, which
are eventually carried into the chromatographic columns by helium. Individual VOCs separated
in the columns are eventually detected by FID systems. Formaldehyde (HCHO) was
continuously measured by a Hantzsch fluorescence technique (AL4201, Aerolaser GmbH.,
GER), which is based on fluorometric Hantzsch reaction in the liquid phase, requiring the
quantitative transfer of HCHO from gas phase to liquid phase. A Hantzsch reagent
(acetylacetone) was used in this instrument.
**2.2 Observation-based model**
In this study, a zero-dimensional (0-D) box model (F0AM) (Wolfe et al., 2016b) based on
the University of Washington Chemical Model (UWCM) was used to simulate the atmospheric
chemical processes. Dry deposition and atmospheric dilution were considered in this model.
The Master Chemical Mechanism (MCM) v3.3.1 with more than 5,800 chemical species and
17,000 reactions was  used in this study to enable a detailed description of the complex
chemical reactions. In addition to gas-phase reactions, several heterogeneous processes
including the uptake of $HO_2$, $N_2O_5$ and HCHO on aerosol surface as well as heterogeneous
sources of nitrous acid (HONO) were considered in our simulation. The rate constants and
uptake coefficient of these reactions were obtained from the study of Riedel et al. (2014), Xue
et al. (2014) and Li et al. (2014). Since key parameters such as aerosol surface area ($S_A$) and
particle diameter (r) were not measured, an average $S_A$ (640 $nm^2/cm^3$) was adopted from the
field campaign in Shanghai (Wang et al., (2014)).
**Table 2. Heterogeneous reactions and associated rate constants used in the OBM model.**

| Reactions | Reaction rate constant | Reference |
|---|---|---|
| $N_2O_5 \rightarrow CLNO_2 + HNO3$ | $\gamma \omega S_A/4$ (for CLNO$_2$ formation) $(2 - \emptyset)\gamma \omega S_A/4$ (for HNO$_3$ formation) | Riedel et al. (2014) |
| $NO_2 \rightarrow HONO$ | $k_g = \dfrac{1}{8} \times \omega \gamma_g (\dfrac{S}{V})$ $k_a = \dfrac{1}{4} \omega \gamma_a S_A$ | Xue et al. (2014) |
| $HO_2 \rightarrow products$ | $k = (\dfrac{r}{D_g} + \dfrac{4}{\gamma} \omega)^{-1} S_A$ | Xue et al. (2014) |
| $HCHO \rightarrow products1$ | $k = \dfrac{1}{4} \omega \gamma S_A$ | Li et al. (2014) |

γ= uptake coefficient for the given reactant with aerosol surface area; ϕ = product yield; ω=mean

molecular speed of the given reactant (m/s); $S_A$=RH corrected aerosol surface area concentration

($nm^2/cm^3$); r=surface-weighted particle radius.


Photolysis frequencies (*J* values) were calculated by a trigonometric parameterization
based on solar zenith angle (SZA):

$$J = Icos(SZA)^m \exp(-nsec(SZA)) \qquad (1)$$

where *I*, *m* and *n* are constants unique to each photolysis reaction, derived from least-
squares fits to *J* values computed with fixed solar spectra and literature cross-section and
quantum yields (Wolfe et al., 2016b). Hourly average concentrations of speciated VOCs
(except HCHO), NO, NO$_2$, CO and meteorological parameters (such as T, RH and P) were
used to constrain the F0AM model. Since nitrous acid (HONO) was not measured during our
observation, it was fixed as 2% of the observed $NO_2$ concentration. This constant ratio is well
observed in different field studies and performed well in previous box model studies (Tan et
al., 2019). Before each simulation, the model was run 3 days as spin up to reach a steady state
for unmeasured species (e.g., OH and $NO_3$ radicals).
The comparison of simulated and observed $O_3$ and HCHO concentrations are shown in
Figure S1 and Figure S2. The index of agreement (IOA), mean bias (MB) and normalized mean
bias (NMB) are used to evaluate the model performance. These three parameters can be
calculated by Equation (2) to (4), where $S_i$, $O_i$, and $\overline{O}$ are the simulated, observed, and average
observed value of the target compound. In this study, the IOA, MB and NMB of $O_3$ was 0.90,
0.76 and 10%, respectively. These results suggest that the model can reasonably reproduce the
variations of $O_3$ and could be used for further analysis. As for HCHO, the IOA, MB, and NMB
was 0.74, 2.43 and 48%, respectively. In general, the model overestimated HCHO
concentration, especially on July 29 and July 30. According to previous studies, the
inconsistency between simulated and observed HCHO is attributed to uncertainties in the
treatment of dry deposition, faster vertical transport, uptake of HCHO and fresh emissions of
VOCs precursors (Li et al., 2014). In addition, primary HCHO sources can contribute up to 76%
of total HCHO concentration in urban areas (Li et al., 2010). However, due to the lack of
primary HCHO sources for areas around DSL, primary HCHO emissions were not included in
our model. Although there exists some bias, the model results still provide valuable information
of secondary formation of HCHO at suburban areas. To assess the reliability of model results
without OH observation, we compared the OBM-simulated OH concentration with that
calculated using the ratio of ethylbenzene (E) and m,p-xylene (X) that share common emission
sources but with different reactivity with OH radicals (shown in Equation (5)~(8)):

$$IOA = 1 - \frac{\sum(S_i - O_i)^2}{\sum(|S_i - \bar{O}| + |O_i - \bar{O}|)^2} \tag{2}$$

$$MB = \frac{\sum(S_i - O_i)}{N} \tag{3}$$

$$NMB = \frac{\sum(S_i - O_i)}{\sum O_i} \times 100\% \tag{4}$$

$$Ethylbenzene + OH \rightarrow products$$
$$k_{Ethylbenzene,OH} = 7.0 \times 10^{-12}(cm^3\ molecule^{-1}\ s^{-1}) \tag{5}$$

$$m,p - Xylene + OH \rightarrow products$$
$$k_{m,p-Xylene,OH} = 1.89 \times 10^{-11}(cm^3\ molecule^{-1}\ s^{-1}) \tag{6}$$

$$[X]_t = [X]_0 \times e^{-[OH]\times k_{X,OH}\times t} \times f_{d,B} \tag{7}$$

$$[OH]_{\frac{E}{X}} = \frac{1}{t \times \left(k_{E,OH} - k_{X,OH}\right)} \times [ln\left(\frac{[E]}{[X]}\right)_0 - ln(\frac{[E]}{[X]})_t] \tag{8}$$

where $[X]_0$ and $[X]_t$ are the mixing ratio of X at the initial time and after transport time t. $k_{X,OH}$
is the temperature dependent reaction rate coefficient of m,p-xylene with OH, which was taken
from the IUPAC database (http://iupac.pole-ether.fr/), whereas $f_{d,B}$ is the dilution factor of m,p-
xylene in the atmosphere. In this study, we assume that the rates of turbulent mixing and
horizontal convection are similar for E and X. Therefore, during the transport time Δt, the
dilution factor of E and X are the same. Therefore, rearranging Equation (7) and extending this
analysis to E and X will yield Equation (8), where $[OH]_{E/X}$ is the estimated regional mixing
ratio of OH by ethylbenzene and m, p-xylene ratio. The calculated average regional
concentrations of OH ($8.39 \pm 5.11 \times 10^6$ molecules cm$^{-3}$) was in the same magnitude of the
OBM-simulated result ($4.59 \pm 5.11 \times 10^6$ molecules cm$^{-3}$), suggesting that the OBM-simulated
radical concentration is reliable.

186        To quantify the changes of atmospheric oxidative capacity (AOC) in response to isoprene

chemistry, two parallel scenarios (S0 and S1) were conducted with isoprene chemistry disabled
in S1. In both cases, identical chemical mechanism and meteorological conditions were used
to drive the model simulations. A comparative analysis of the scenarios revealed the impact of
isoprene chemistry on AOC and secondary formation of $O_3$ and HCHO.

## 3. Results and discussions

### 3.1 Overview of the observations

193        To investigate the impact of local chemistry on ozone formation and avoid the influence

of emission transportation, five days under stagnant condition (with daily average wind speed
less than 2m/s and maximum daily 8-h average (MDA8) $O_3$ concentration >75 ppb) were
identified as typical local chemistry cases. Figure 2 shows the time series of observed
meteorological parameters (P, T, and RH), trace gases (NO, $NO_2$ and $O_3$), isoprene and HCHO
on selected days. During those episodes, the air masses reaching the site were mainly from
southeast and southwest (Figure 2). The weak wind was not conductive to the regional
transportation of air pollutants. The observed $O_3$, $NO_2$, NO, CO, and TVOC ranged from 1.40
to 155.40 ppbv ($52.72 \pm 44.43$ ppbv, average value, the same below), 5.36 to 57.95 ppbv (21.58
$\pm 12.88$ ppbv), 0.75 to 54.51 ppbv ($5.40 \pm 8.13$ ppbv), 400 to 960 ppbv ($597 \pm 153$ ppbv), and
2.34 to 20.33 ppbv ($7.28 \pm 4.32$ ppbv) respectively. During the five episodes, the average
concentration of alkanes ($13.97 \pm 9.12$ ppbv), alkenes ($3.27 \pm 2.31$ ppbv) and aromatics (4.93
$\pm 2.69$ ppbv) was about 53%, 18%, and 50% respectively, higher than that of the whole
observation period. The conditional probability function (CPF) is applied to reveal the
relationship between high $O_3$ concentrations and wind (Figure 3). The detailed description of
CPF can be found in the supplementary information (Text S1). The results suggest that high
$O_3$ concentrations (>131 ppb) was usually observed when the site was influenced by weak
south wind. This implies that the high $O_3$ was mostly formed locally. Although this site is
distant from urban areas, high levels of NO were found during early morning, emanating
particularly from nearby heavy-duty vehicle emissions. As for $NO_2$, only one peak was found
at dusk. This was in contrast with previous results in urban areas (Zhang et al., 2019). It is
worth noting that $NO_2$ and $O_3$ concentrations were high even during nighttime, suggesting that
the AOC remained high at nighttime. It should also be noted that, flat CO pattern was found
during morning when $NOx$ peaks were observed. This inconformity can be attributed to the
coarse resolution of CO analyzer (about 80 ppbv) and CO emission source (mainly gasoline
vehicles in terms of vehicle exhaust) while $NOx$ is mainly emitted by heavy-duty vehicle
exhausts. Therefore, since DSL site is far from urban area, it is unlikely to have gasoline
vehicles in early morning. On the contrary, there are sometimes heavy-duty trucks passing by,
causing peaks of NO in early morning.
The daily average isoprene concentration was $0.37 \pm 0.36$ ppbv, and is comparable to the
observations by Gong et al. (2018) at a forested mountaintop site ($0.287 \pm 0.032$ ppbv). To
estimate the influence of isoprene on atmospheric oxidation capability, we adopted the
approach by Zhu et al. (2020) to calculate the OH reactivity ($k_{OH}$). Results suggested that
isoprene, accounting for ~19% of the total $k_{OH}$, was the most significant VOC species with
respect to $k_{OH}$ ($0.89 \pm 0.44$ s-1). This indicates the significant role of isoprene in the
photochemistry of a suburban area. The average HCHO was $5.01 \pm 3.80$ ppbv, which was ~2
times of that observed at a rural site of Hong Kong (Yang et al., 2020). It is worth noting that
HCHO could reach an average of 18.69 ppbv at midday.
Based on explicit calculation, the total concentration of OVOC was obtained. However,
due to the complexity of OVOC formation, which could have hundreds of precursors for just
one OVOC specie and the complex chain reactions converting VOCs to OVOCs, it is difficult
to derive an accurate relationship between VOCs and OVOCs. But since VOCs were mainly
oxidized by OH and $O_3$ during daytime, in this study, we chose multi-linear regression model
(Eq.(9) ) to roughly explore the relationship between VOCs and simulated OVOCs.

$$[OVOC] = \beta_0 + \beta_1[Alkane] + \beta_2[Alkene] + \beta_3[Aromatic] + \beta_4[OH] + \beta_5[O_3] \quad (9)$$

where $\beta_0$, $\beta_1$, $\beta_2$, $\beta_3$, $\beta_4$, and $\beta_5$ are the coefficients from linear regression; [OVOC] and [OH]
are the predicted concentration of OVOC and OH, respectively; [Alkane], [Alkene],
[Aromatic], [$O_3$] are the observed concentration of alkanes, alkenes, aromatics, and $O_3$,
respectively. The Sig value and statistical reliability criteria (R) was 0.000 and 0.853 (shown
in Table S2 and Figure S3), respectively, indicating that the linear relationship represented by
equations (9) is statistically reliable. Similarly, the $\beta_1$, $\beta_2$, $\beta_3$ was 0.027, 0.623, and 0.820,
respectively, suggesting that alkenes and aromatics are significant for the simulated OVOC
concentration.

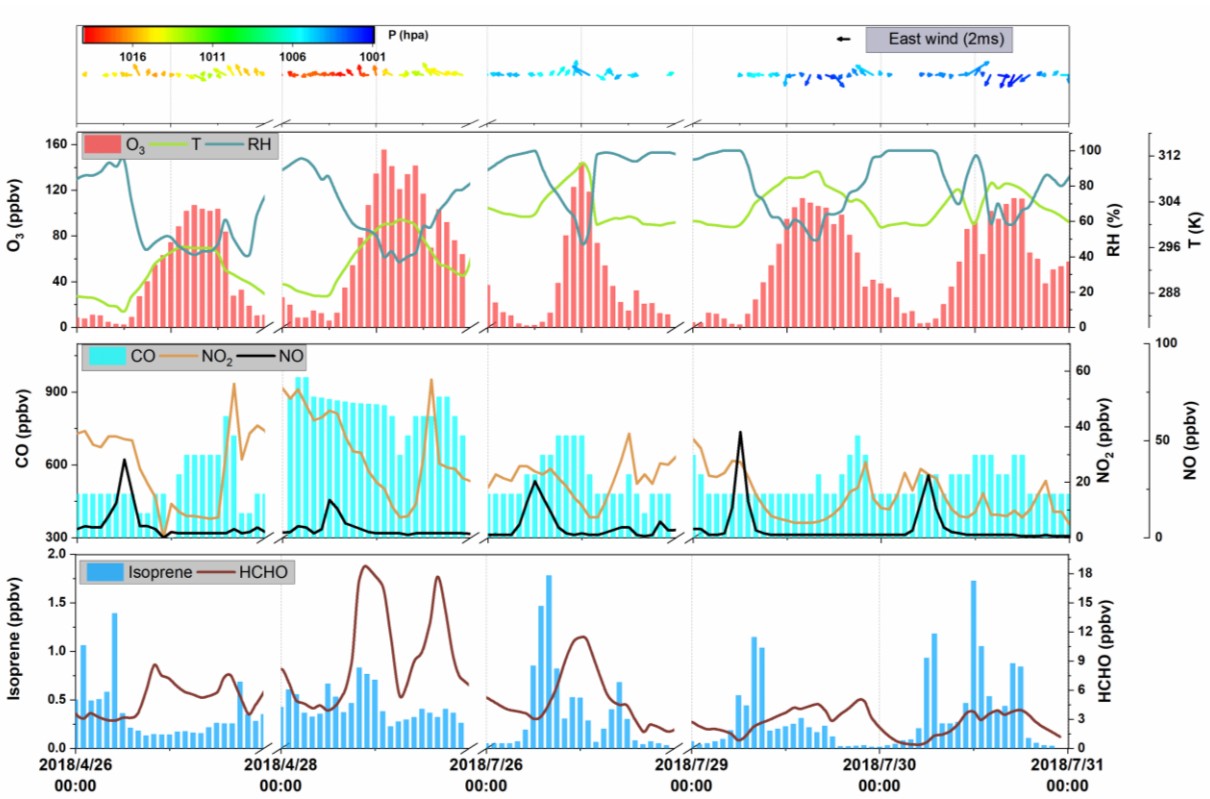


**Figure 2. Time series of hourly averages for $O_3$, CO, NO, $NO_2$, isoprene, HCHO, and meteorological**
**parameters.**

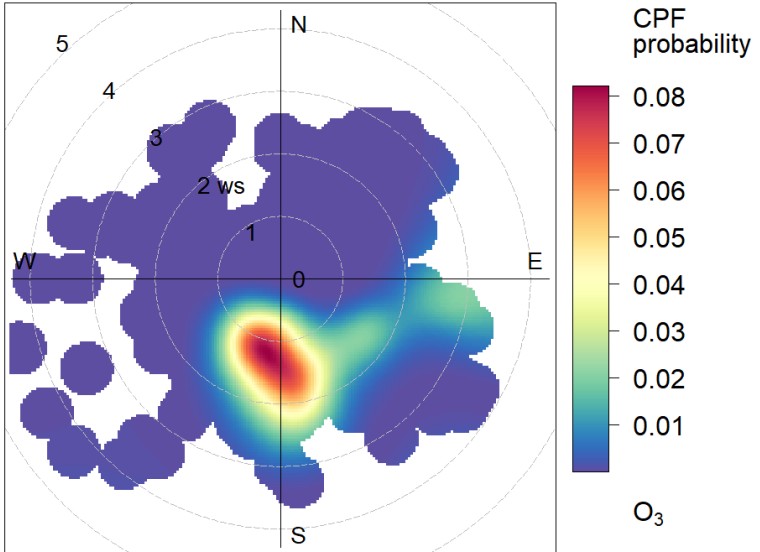

**CPF at the 95th percentile (=131)**

**Figure 3. CPF polar plot of O$_3$ at DSL station.**

### 3.2 Simulated concentrations of radicals

Figure 4 shows the simulated average daytime variation of major radicals in the base scenario (S0). It should be noted that, the discussion below is limited to local conditions (cases with average wind speed lower than 2m/s), since transportation of emissions are not considered in the 0-dimensional model. The daily average simulated concentration of OH, HO$_2$, RO$_2$, and NO$_3$ was $4.88\times10^6$, $3.49\times10^8$, $0.31\times10^{9,}$ and $0.31\times10^8$ molecules cm$^{-3}$, respectively. The simulated daily average OH concentration is comparable to a summertime simulation in Beijing ($9\times10^6$ molecules cm$^{-3}$) (Liu et al., 2019) and at a suburban site in Hong Kong in 2013 ($1.5\pm0.2\times10^6$ molecules cm$^{-3}$) (Xue et al., 2016). In addition, the average simulated daytime OH concentration was ~33% lower than that simulated at a forested mountaintop site in southern China (Gong et al., 2018). To verify the performance of OBM model, regional daytime mixing ratios of OH were also calculated by a parameterization method using measured ethylbenzene and *m,p*-xylene ratios. The calculated average regional concentrations of OH ($8.39\pm5.11\times10^6$ molecules cm$^{-3}$) was in the same magnitude of the OBM-simulated

result ($4.88 \pm 5.11 \times 10^6$ molecules cm$^{-3}$), suggesting that the OBM-simulated radical
concentration is reliable. Furthermore, at the DSL site, the simulated maximum HO$_2$
concentration ($6.19 \times 10^8$ molecules cm$^{-3}$) was close to that reported in Beijing ($6.8 \times 10^8$
molecules cm$^{-3}$) (Liu et al., 2012), but was ~32% higher than that in Wuhan ($4.7 \times 10^8$ molecules
cm$^{-3}$) (Zhu et al., 2020a). Due to high reactivity of RO$_2$ and high concentration of HO$_x$, RO$_2$
kept low level during daytime. As for NO$_3$, it can be quickly decomposed during daytime,
leading to the negligible concentration in the daytime.

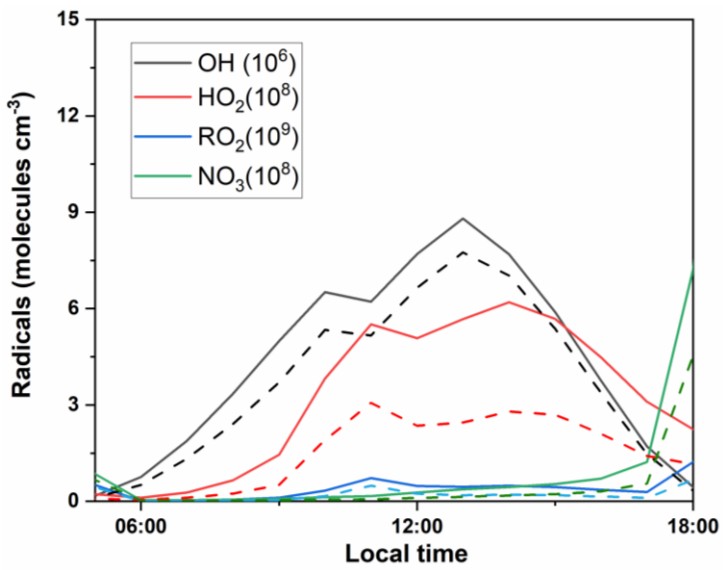


**Figure 4. Simulated average daytime variation of OH, HO$_2$, RO$_2$ and NO$_3$ in S0 (solid lines) and S1 (dash**
**lines).**

## 3.3 Recycling of RO*x* radicals

Figure 5(A) shows the primary sources of RO*x* in S0 and its detailed daytime budget.
Minor RO*x* sources, e.g. ozonolysis of alkenes, are not shown in the figure. Photolysis of O$_3$
was the predominant primary source of OH, with a daytime mean production rate of 0.50 ppbv
h$^{-1}$, which was comparable to that found by Liu et al. (2012) in Beijing, but was 0.40 ppbv h$^{-1}$
lower than the result reported by Xue et al. (2016). Similarly, another important OH source is
the photolysis of HONO, contributing 0.32 ppbv h$^{-1}$ of daytime OH production in our
simulation. This result is much lower than other related studies (Liu et al. (2019) and Xue et al.
(2016)), possibly due to the excessive constrain on HONO (since HONO was not directly
monitored) during our experiment.
Sensitivity analysis was conducted to quantify the influence of different $HONO/NO_2$ ratio
on radical recycling (Text S2, Figure S4 and Table S1). As expected, a lower $HONO/NO_2$ ratio
leads to a lower HONO concentration, and subsequently less OH generation from the
photolysis of HONO. The sensitivity analysis shows that when $HONO/NO_2$ ratio is 0.005, the
daytime OH level could decrease by 15.28%. Conversely, a higher $HONO/NO_2$ (e.g., 0.04) can
promote OH concentration by 14.08%. This result illustrates the importance of HONO
photolysis in the generation of OH, and therefore simultaneous ambient measurements of
HONO is highly recommended for future analysis of local radical recycling. Regarding $HO_2$,
the photolysis of OVOC (excluding HCHO) is the predominant source with a daytime mean
production rate of 0.65 ppbv $h^{-1}$ and maxima of 0.92 ppbv $h^{-1}$, which is comparable to Xue et
al. (2016). The photolysis of HCHO can also contribute 0.48 ppbv $h^{-1}$ to the daytime production
of $HO_2$, which is close to the results of Xue et al. (2016). As for $RO_2$, the photolysis of OVOC
was the largest source (0.57 ppbv $h^{-1}$), which was relatively lower than the results found at an
urban site (Liu et al., 2012). Therefore, regarding RO$x$ in DSL site, the daytime primary radical
production was dominated by the photolysis of OVOC (except for HCHO), followed by the
photolysis of HCHO and $O_3$. However, the photolysis of HONO can become the overwhelming
RO$x$ source around sunrise, which suggests that HONO can be an important OH reservoir
species at night. Summing up all the sources of RO$x$ gives a total primary daytime RO$x$
production rate of 2.55 ppbv $h^{-1}$ (0.84 ppbv $h^{-1}$ for OH, 1.14 ppbv $h^{-1}$ for $HO_2$, and 0.57 ppbv
$h^{-1}$ for $RO_2$), which was 61~69% lower than those in Beijing (6.6 ppbv $h^{-1}$, Liu et al. (2012))
and Hong Kong (8.11 ppbv $h^{-1}$, Xue et al. (2016)), indicating that the recycling of RO$x$ in
Beijing and Hong Kong could be much reactive.
RO$x$ radicals are ultimately removed from the atmosphere via deposition of radical
reservoir species, e.g. $H_2O_2$, $HNO_3$, and ROOH (Liu et al., 2012). The terminate processes of
RO$x$ was dominated by their reactions with NO$x$. Specifically in this study, the reaction of
OH+$NO_2$, $RO_2$+$NO_2$, $RO_2$+NO, forming $HNO_3$, $RO_2NO_2$, and $RONO_2$, accounted for 2.42,
0.56, and 0.41 ppbv h$^{-1}$ of the daytime RO$x$ radical sink, respectively. This is consistent with
the understanding that reactions with NO$x$ usually dominate the radical sink under high NO$x$
environments (Xue et al., 2016; Liu et al., 2012). In addition, $RONO_2$ and $RO_2NO_2$ could in
turn react with OH, leading to 0.41 ppbv h$^{-1}$ of daytime OH sinks (Figure 6). Summing up the
primary sources and sinks gives a negative value of net RO$x$ production, suggesting that the
RO$x$ was in a stage of gradual depletion.

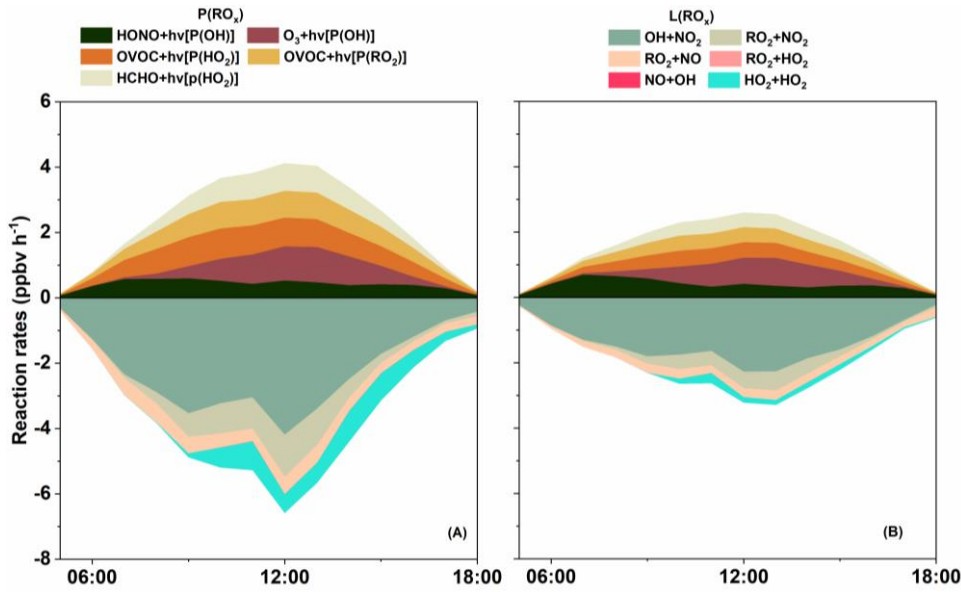

**Figure 5. Simulated primary daytime sources and sink of RO$x$ in S0 (A) and S1 (B).**
Furthermore, the daytime (6:00-18:00) average budget of RO$x$ is shown in Figure 6.
Evidently, the production of OH was dominated by the reaction of $HO_2$+NO (8.29 ppbv h$^{-1}$)
in RO$x$ recycling, whereas $RO_2$ was produced by the reaction of OH with OVOC (3.02 ppbv
h$^{-1}$), alkyl (RH) (1.21 ppbv h$^{-1}$), and peroxides (0.14 ppbv h$^{-1}$). Besides, the reaction of
$RO_2$+NO can result in strong production of RO (3.87 ppbv h$^{-1}$). Moreover, the reaction of RO
and $O_2$ was the major contributor to $HO_2$ production, followed by the reaction of OH with CO
(1.89 ppbv h$^{-1}$), OVOC (1.59 ppbv h$^{-1}$), and RH (0.15 ppbv h$^{-1}$). It is worth noting that the top
two fast reactions within the recycling of ROx (HO$_2$+NO and RO$_2$+NO) were related to NOx.
As mentioned in the study of Liu et al. (2012), this result could be mainly due to the abundance
of NO (e.g. ~50 ppbv in the morning). Obviously, these recycling processes dominate the total
production of OH, HO$_2$ and RO$_2$ radicals. As suggested in the study of Xue et al. (2016) and
Liu et al. (2012), the radical propagation is efficient and enhances the effect of the newly
produced radicals in the polluted atmospheres with the co-existence of abundant NOx and
VOCs.

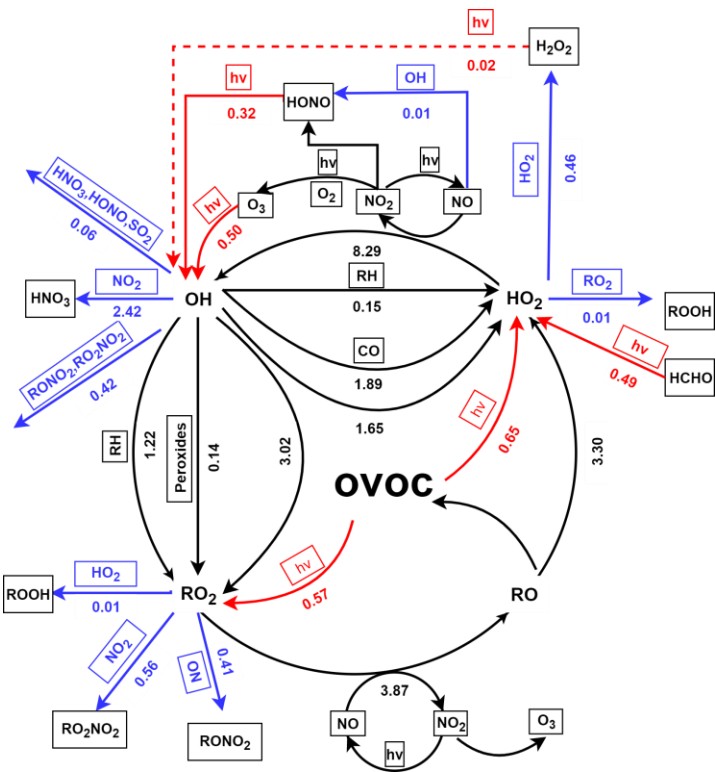


**Figure 6. Summary of daytime (06:00-18:00) average budgets of ROx radicals (in ppbv h$^{-1}$). Primary ROx**
**sources and sinks are in red and blue, respectively, and the black lines represent the processes in ROx**
**and NOx recycling.**
**3.4 Formation and sink of O$_3$**
Figure 7 illustrates the diurnal variation of simulated O$_3$ concentration, net production rate
(including the formation and sink pathways) in S0. In the troposphere, O$_3$ is formed via the
reactions of NO with peroxy radicals (e.g. HO$_2$ and RO$_2$) (Liu et al., 2012; Xue et al., 2016;
Zhu et al., 2020a). Consequently, the daytime reaction of $HO_2+NO$ and $RO_2+NO$ contributed
an average of 9.34 and 8.52 ppbv $h^{-1}$ of the $O_3$ produced. Coincidentally, the maximum rate of
$HO_2+NO$ (15.36 ppbv $h^{-1}$) and $RO_2+NO$ (13.26 ppbv $h^{-1}$) both occurred at 13:00 LST. Our
results reveal a total daytime production rate of $O_3$ ($P(O_3)$: the sum of $HO_2+NO$ and $RO_2+NO$)
at 17.86 ppbv $h^{-1}$, which is in line with related study in Beijing (32 ppbv $h^{-1}$, Liu et al. (2012))
and Hong Kong (6.7 ppbv $h^{-1}$, Liu et al. (2019)).
Due to the fast cycling of both $O_3$ and $NO_2$, the sink of $O_3$ resulted from several reactions
leading to the destruction of $O_3$ and $NO_2$. In our case, the reaction of $NO_2+OH$ is the
predominant scavenging pathway of $O_3$, with an average daytime reaction rate of 1.89 ppbv $h^{-1}$
(49%, percentage of the total $O_3$ sink rate.). This is comparable to the study of Liu et al. (2012
and 2019). In addition, the reaction of $RO_2+NO_2$ was the second contributor to $O_3$ sink with a
mean contribution of 0.62 ppbv $h^{-1}$ (16%). Other pathways, e.g. photolysis of $O_3$, ozonolysis
of alkenes, and $O_3+HO_2$, altogether contributed 1.11 ppbv $h^{-1}$ of the total daytime $O_3$ sink rate.
Also, the daytime mean $L(O_3)$ was 3.87 ppbv $h^{-1}$, which was ~22% of $P(O_3)$, suggesting that
$O_3$ could efficiently accumulate during daytime. The net production of $O_3$ ($P(O_3)-L(O_3)$) is also
shown in Figure 7. Our results reveal a maximum $O_3$ concentration at around 16:00 LST, which
was also observed in other suburban sites (Zong et al., 2018; Zhang et al., 2019).
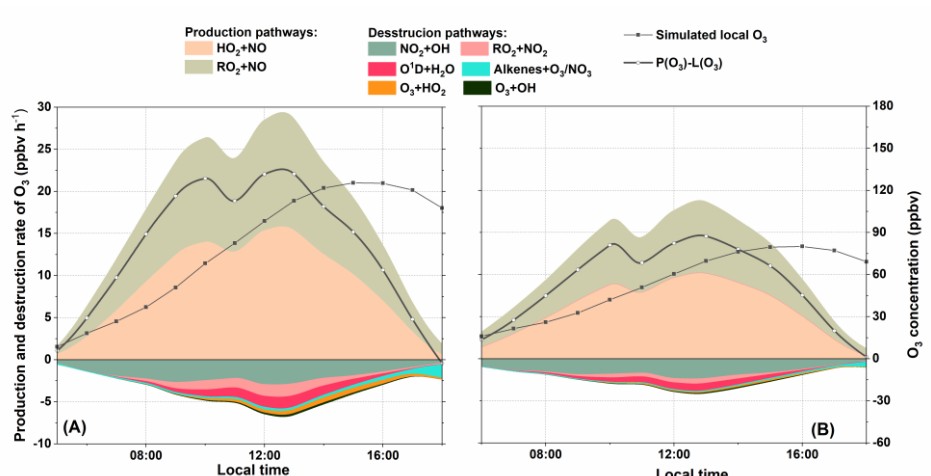
**Figure 7. Simulated average diurnal profiles of $O_3$ formation and sink rates (ppbv $h^{-1}$) in S0 (A) and S1**
**(B).**

**3.5 Formation and sink of HCHO**

As aforementioned, high levels of HCHO was observed at DSL. Figure 8 (A) shows the production and sink pathways of HCHO in S0. In this study, the local HCHO formation was dominated by the reaction of $RO+O_2$, accounting for ~90% of the total production rate. Further classification of $RO+O_2$ pathway suggested that the oxidation of $CH_3O$ made a significant contribution of ~47%, followed by RO (from isoprene) + $O_2$ reaction (12%) and RO (from aromatics) + $O_2$ reaction (~11%). This result is comparable to the study of Yang et al. (2020; 2018). During the day, isoprene is the most important VOC species in the production of HCHO with a mean rate of 0.48 ppbv $h^{-1}$. As stated earlier, the study site is surrounded by dense vegetation, which provides abundant biogenic isoprene. As a result, over 90% of the daytime isoprene was oxidized by OH radicals (Figure S5). Based on the MCMv3.3.1, several $RO_2$ species (e.g. ISOP34O2, ISOPDO2, ISOPCO2, CISOPAO2, ISOPAO2) can be generated during the OH-initiated degradation process of isoprene (Jenkin et al., 2015). For instance, with the presence of NO, isoprene-originated $RO_2$ can transfer into RO (e.g. ISOPDO, ISOP34O, ISOPAO). The subsequent degradation processes of isoprene-related RO, especially ISOP34O, ISOPDO, ISOPAO and ISOPBO, are closely related to the formation of HCHO (Jenkin et al., 2015). In other sources of HCHO, such as the reaction between VOC and $O_3$, photolysis of OVOC and the reaction of OVOC+OH only contributed small amount of the total production rate during whole day.

In this study, it is noteworthy that the two dominant pathways for HCHO depletion were the photolysis of HCHO (~52%) and the reaction of HCHO+OH (~48%). On the other hand, the net HCHO production rate (equals to P(HCHO) + L(HCHO)) as shown in Figure 8. It is evident that after sunrise, the net production rate of HCHO rose gradually to a peak of ~1.6 ppbv $h^{-1}$ at 8:00 (similar to the results by Yang et al. (2018)). Thereafter, at around 12:00 LST, net(HCHO) dropped to ~0 ppbv $h^{-1}$, which was roughly consistent with our observation,

showing the HCHO peak occurs at around 12:00. In addition, a negative net(HCHO) was
exhibited between 13:00 and 14:00, . Although the reaction of $RO+O_2$ quickly produced HCHO
in the afternoon, the depletion pathways, especially the photolysis of HCHO, became more
competitive, leading to the net reduction of HCHO. This also indicates that strong
photochemical reactions do not monotonously profit the accumulation of HCHO, it can also
constrain high HCHO levels in certain situations. After 14:00, the photolysis of HCHO dropped
rapidly and the net depletion of HCHO back to ~0 ppbv $h^{-1}$ at around 15:00. The daytime net
HCHO production rate was 0.70 ppbv $h^{-1}$, which was comparable to result of Yang et al. (2018).
The above analysis indicates that the photolysis of OVOC, HCHO, $O_3$ and HONO was
the primary source of RO$x$, which offers high oxidizing environment for the degradation of
VOCs. As a typical by-product in the degradation of several VOCs, HCHO can be quickly
formatted during the day. The insight into the detailed photochemical processes shows the
important role of isoprene in the formation of HCHO.

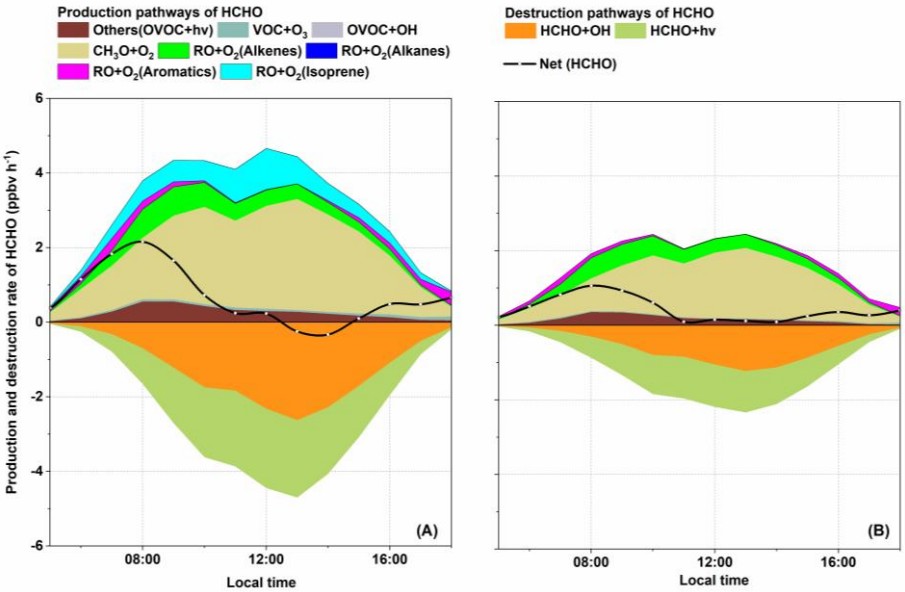


**Figure 8. Simulated average daytime profiles of net rate (net (HCHO)), breakdown HCHO production**
**rate and sink rate (ppbv $h^{-1}$) in S0 (A) and S1 (B).**

**3.6 Impacts of isoprene chemistry on photochemistry**

**3.6.1 Impact on RO*x* budget**

To compare the importance of isoprene and other abundant VOCs in local chemistry at DSL site, sensitivity analysis was conducted for the modelled $O_3$, HCHO, and OH concentrations without the input of active VOCs (toluene, ethylene, ethylbenzene, ethane, acetylene, xylene, propene, and isoprene). Results suggested that, although the average isoprene concentration was only $0.37 \pm 0.36$ ppbv, cutting isoprene input can lead to obvious drop in simulated $O_3$, HCHO, and OH, which was comparable to that of cutting EXT and alkenes, indicating the significant role of isoprene in local photochemical processes (Figure S6). In addition, the degradation of isoprene is closely linked to the cycling of RO*x*. To roughly explain the impact of isoprene chemistry on RO*x* budget, we carried out a parallel simulation (S1) where isoprene chemistry is disabled (Figure 9). The diurnal variation of OH, $HO_2$, $RO_2$ and $NO_3$ in S1 is also shown in Figure 4 (B) which clearly suggests the decline in RO*x* and $NO_3$ without isoprene input. To investigate the underlying causes, we calculated the production rate of RO*x* (P(RO*x*)) and loss rate of RO*x* (L(RO*x*)) in S1, respectively (Figure 5 (B)). Comparative analysis revealed a decreasing trend for most of the reaction rates in P(RO*x*) and L(RO*x*) in S1. This strongly indicates that the absence of isoprene slows down the RO*x* recycling. Generally, considering that the photolysis of OVOC (0.67 ppbv $h^{-1}$) was still the predominant primary source of RO*x*, but without isoprene, the photolysis rate of OVOC decreased by 0.49 ppbv $h^{-1}$. Moreover, the total production and depletion rate of OH dropped to 6.96 and 7.51 ppbv $h^{-1}$, respectively. Although the absence of isoprene could reduce the consumption of OH, the OH concentration would be reduced by ~16% compared to S0, suggesting that the amount of OH produced via isoprene chemistry is large enough to compensate for the shift from OH to peroxy radicals in the RO*x* family. As for $RO_2$, the daytime production and sink rate falls to 3.25 and 3.34 ppbv $h^{-1}$, respectively. This means the

concentration of RO₂ would be in a stage of gradual decrease. In addition, the absence of
isoprene could also reduce RO₂ concentration by ~20%, suggesting that isoprene was an
important source of RO₂ at DSL site. As for HO₂, drastic decrease of ~53% was found in S1.
The above-mentioned decrease in RO$x$ obviously could not be explained solely by the removal
of isoprene-related radicals. Sensitivity assessment of the model results shows that OVOC
concentrations decreased drastically (~41%) after cutting isoprene (e.g. ~37% decrease in
formaldehyde, ~65% decrease in methylglyoxal, ~51% decrease in glyoxal, ~100% decrease
in methacrolein (MACR), and ~100% decrease in methyl vinyl ketone (MVK)). The decrease
in OVOC can further pull-down substantial amount of primary RO₂ and HO₂ (Figure 6 and
Figure 9). It is interesting to note that, taking away isoprene also causes a drop of NO₃ (~23%).
This result can be attributed to the decrease of secondary production of O₃ (~35%), which can
further reduce the formation of NO₃ especially at night.

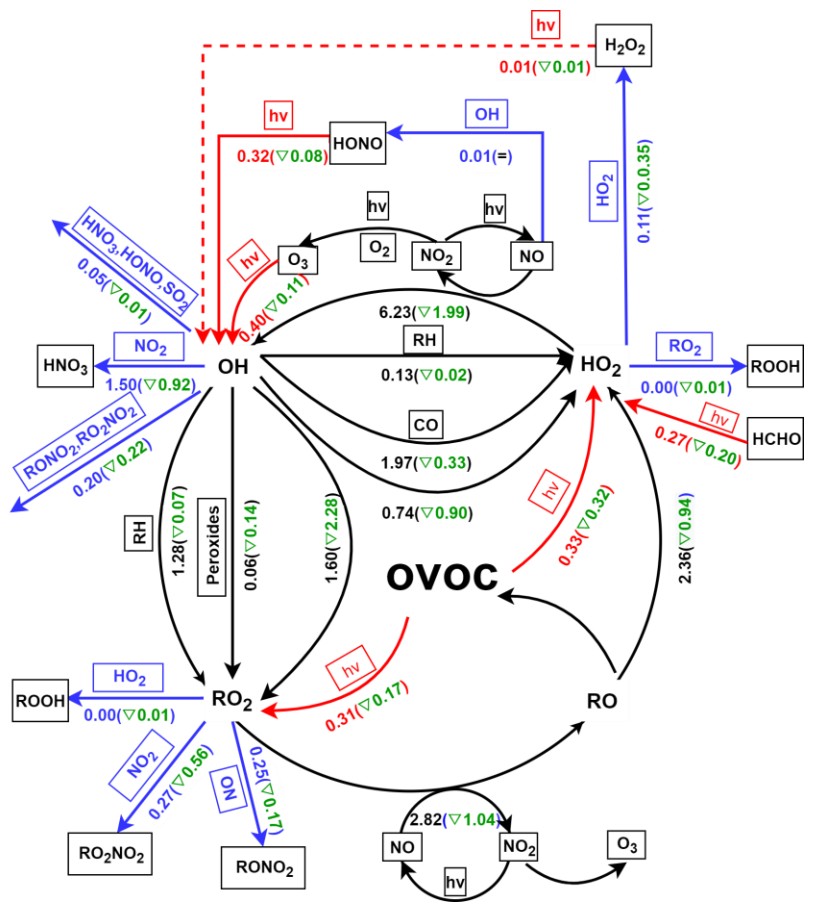


### 3.6.2 Impact on $O_3$ formation

To investigate the detailed impact of isoprene on $O_3$ formation, the production and sink pathways of $O_3$ in S1 was also quantified (see Figure 7 (B)). Notably, the simulated maximum and daily average $O_3$ dropped to 84.95 and 41.23 ppbv, respectively, which is ~35% and ~34% lower than that in S0. By comparing S1 and S0, the absence of isoprene can reduce all the production and sink pathways of $O_3$. For example, the rate of the two major production pathways of $O_3$ ($HO_2$+NO and $RO_2$+NO) decreased by ~37% and ~45%, respectively. This can be attributed to the drop in the concentration of $HO_2$ and $RO_2$ racial in S1. Similarly, the absence of isoprene in $O_3$ depletion caused a decrease of 0.31 ppbv h$^{-1}$ in the reaction rate of alkene+$O_3$/$NO_3$, followed by $RO_2$+$NO_2$ (0.22 ppbv h$^{-1}$) and $NO_2$+OH (0.265 ppbv h$^{-1}$). Apparently, the absence of isoprene will reduce the total concentrations of alkenes and can further lead to the decrease of $RO_2$ and OH level, which ultimately slows down the depletion pathways of $O_3$. Eventually, the absence of isoprene caused a decrease of 5.78 ppbv h$^{-1}$ in the daytime mean net production rate of $O_3$. Hence, isoprene chemistry plays an important role in the local $O_3$ formation at DSL site.

### 3.6.3 Impact on HCHO formation

The analysis of S0 revealed the important role of isoprene, aromatics, and alkenes in the production of HCHO. To investigate the chain effect of isoprene chemistry on HCHO production, the major reactions that dominate the formation and depletion of HCHO in S1 were also analyzed by OBM model (Figure 8 (B)). Comparisons between S0 and S1 shows that the daily average HCHO decreased by 2.90 ppbv (~39%) when isoprene chemistry is cut off. Obviously, the drop in HCHO concentration cannot be solely illustrated by the absence of RO (from isoprene). As aforementioned, the absence of isoprene slows down the recycling of RO*x*

and can further lead to decrease in ROx concentration. Based on the OBM analysis, the
concentration of $CH_3O$, RO (from aromatics), RO (from alkanes), and RO (from alkenes)
decreased by $2.70 \times 10^2$ molecule $cm^{-3}$, $1.59 \times 10^5$ molecule $cm^{-3}$, $3.35 \times 10^1$ molecule $cm^{-3}$, and
3.44 molecule $cm^{-3}$, respectively. The drop in the HCHO precursor concentrations ultimately
led to decrease in the daytime reaction rate of $CH_3O + O_2$, RO (from alkenes) $+ O_2$, and RO
(from aromatics) $+ O_2$ by 0.66 ppbv $h^{-1}$ (~36%), 0.06 ppbv $h^{-1}$ (~16%), and 0.06 ppbv $h^{-1}$
(~40%), respectively. The total daytime formation rate of HCHO dropped to 1.71 ppbv $h^{-1}$ from
1.66 ppbv $h^{-1}$ (~49%) lower than that in S0. As a result of the lower HCHO and OH
concentration in S1, the daily mean depletion rate of HCHO decreased by 1.25 ppbv $h^{-1}$ (~49%).
Ultimately, the absence of isoprene pulled down the daily average HCHO level by 1.61ppbv
(~36%).
**3.7 Uncertainty analysis**
Due to limitations in the observations, several issues should be noted in the application of
the OBM model to evaluate the local chemistry in the present study. Firstly, methane
concentration, which was set to 1850 ppbv based on previous observations, could be an
overestimation or underestimation. Thus, we conducted sensitivity analysis of modelled $O_3$,
OH, and HCHO with different methane values (from 1600 ppbv to 1900 ppbv) (Figure. S7).
The model predicted $O_3$, HCHO, and OH concentration with negligible change under different
$CH_4$ values. Secondly, the photolysis rates directly influence the key photochemical processes
during the day. Since the photolysis rates were not measured during the sampling period, we
also conducted sensitivity analysis by increasing or decreasing the photolysis rates by 20% and
40%. Results showed that the $O_3$, HCHO and OH concentration could increase by 51.14%,
34.52%, and 50.38%, respectively, when photolysis rates were increased by 40% (Figure S8).
On the contrary, when photolysis rates were decreased by 40%, $O_3$, HCHO and OH
concentration decreased by 50.59%, 30.84%, and 47.24%, respectively (Figure. S6). According
to the study by Xu et al. (2013), $NO_2$ concentration measured by the molybdenum oxide
converter technique can be significantly overestimated in areas far away from fresh $NOx$
emission sources. Therefore, OBM simulations with reduced $NO_2$ concentrations were
conducted. The results suggest that decreasing $NO_2$ could increase or decrease of $O_3$, HCHO
and OH concentrations under different scenarios (Figure S9). Overall, decreasing $NO_2$ by 40%
could cause 6.94%, 12.07%, and 6.29% increase in $O_3$, HCHO, and OH concentrations,
respectively. Finally, the total surface area of aerosols was obtained from the study of Wang et
al. (2014) and the uncertainty of this value could directly influence the heterogeneous reactions
in this model. Therefore, we conducted sensitive analysis by using increasing or decreasing SA
value by 40% (Figure S10). The results show that $O_3$, HCHO, and OH concentrations did not
exhibit obvious changes when SA changed. Hence, accurate measurement data of photolysis
rate and $NO_2$ concentration is strongly recommended in further OBM analyses.

## 4. Conclusions

Our observations at a suburban site of the YRD region from April to June in 2018 captured
5 typical local $O_3$ formation episodes. The detailed atmospheric photochemistry during these
episodes were analyzed by a typical 0-D box model on a local scale. Under stagnant conditions,
the photolysis of OVOC served as the predominant primary $ROx$ sources. $ROx$ achieves
efficient recycling with the participation of $NOx$. Influenced by the fast $ROx$ recycling, local
$O_3$ was efficiently produced and accumulated under stagnant conditions. The reactions of RO
radicals with $O_2$ dominate the photochemical formation of HCHO. The higher atmospheric
oxidative capacity lead to fast degradation of VOCs, which can further lead to high levels of
HCHO at the DSL site. Specifically, the degradation of RO radicals (e.g. ISOP34O, ISOPDO,
ISOPAO and ISOPBO) from isoprene oxidation play an important role in the photochemical
production of HCHO. To investigate the role of isoprene in $ROx$ recycle and the formation of
secondary pollutant, a sensitivity scenario without isoprene (S1) input was simulated by OBM
model. By comparing S1 to the standard simulation (S0), we find that isoprene chemistry is
important in local $RO_x$ recycling. The absence of isoprene can obviously decrease the
concentrations of OVOC and the reaction rates in $RO_x$ propagations, and further reduce the
concentrations of radicals (e.g. OH, $HO_2$, $RO_2$). Our results indicate that isoprene chemistry
can strongly influence the formation of $O_3$ and HCHO in the presence of $NO_x$. Therefore,
removing isoprene can slow down the reaction of $HO_2$+NO and $RO_2$+NO by ~37% and ~45%,
respectively, and eventually cause ~34% decrease of $O_3$. As a result of the lower $O_3$
concentration, average concentration of $NO_3$ dropped by 23% in S1. The absence of isoprene
can also lead to the decrease of RO (from isoprene) and $RO_x$ concentration and cause an
obvious drop of HCHO formation (~49%). Furthermore, other biogenic VOCs (BVOCs, such
as terpene and sesquiterpene) can also affect local chemistry via photochemical processes, but
those BVOCs were not able to be synchronously observed. Therefore, future studies should
take into account those BVOCs. Additionally, the uncertainty analysis conducted in this study
indicates the significance of synchronous and accurate observation of photolysis rates and $NO_2$
concentration when using the OBM. Generally, this study underlines the significant role of
isoprene chemistry in radical chemistry, photochemical reactions, and secondary pollutant
formation in the atmosphere of the YRD region and provides insights into secondary pollution
and its formation mechanisms.

*Data availability.* The data that support the results are available from the corresponding author
upon request.

*Authorship contribution.* Kun Zhang: Formal analysis, Methodology, Writing-original draft.
Ling Huang: Writing-review. Qing Li: Formal analysis. Juntao Huo: Formal analysis, Data
curation. Yusen Duan: Formal analysis, Data curation. Yuhang Wang: Writing-review. Elly
Yaluk: Formal analysis. Yangjun Wang: Formal analysis. Qingyan Fu: Formal analysis. Li Li:
Conceptualization, Methodology, Writing-review & editing.

*Competing interest.* The authors declare that they have no known competing financial interests
or personal relationships that could have appeared to influence the word reported in this paper.

*Acknowledgements.* This study is supported by the National Natural Science Foundation of
China (No. 42075144; No.41875161), Shanghai International Science and Technology
Cooperation Fund (No. 19230742500), Shanghai Science and Technology Fund (No.
19DZ1205007), Shanghai Sail Program (NO.19YF1415600), and the National Key Research
and Development Program of China (NO.2018YFC0213600). Y. Wang was supported by the
National Science Foundation. We thank Shanghai Environmental Monitoring Center (SEMC)
for conducting the measurement and sharing the data.


*Financial support.* This study was financially supported by the National Natural Science
Foundation of China (NO. 41875161; NO.42075144), Shanghai International Science and
Technology Cooperation Fund (NO. 19230742500), Shanghai Science and Technology Fund
(No. 19DZ1205007), Shanghai Sail Program (NO.19YF1415600), and the National Key
Research and Development Program of China (NO.2018YFC0213600). Y. Wang was
supported by the National Science Foundation.

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
