# Peer review of "Explicit modelling of isoprene chemical processing in polluted air"

_Atmospheric Chemistry and Physics, 2020_

## Referee Comment (RC1) · Anonymous Referee #1 · 29 Sep 2020

The manuscript "Observations and explicit modeling of isoprene chemical processing in polluted air masses in rural areas of the Yangtze River Delta region: radical cycling and formation of ozone and formaldehyde" by Zhang et al. is a modelling study of isoprene chemistry in a rural region near Shanghai, China. The main focus of the study is the role of isoprene and oxygenated VOCs in the HOx radical cycle, a subject that has been studied elsewhere but not in this particular region. As such, the results in the paper provide an interesting, if not especially surprising, insight in the atmospheric chemical processes of this part of the world. The paper certainly fits in the remit of ACP,

but I have some serious concerns about the methodology that need to be addressed by the authors before the manuscript can be considered for publication.

MAJOR COMMENTS

I find two main issues with the analysis presented in sections 3.2 and 3.3. One is the lack of heterogeneous chemistry in the model. This is likely to impact both the levels and the budget of OH (because of the heterogeneous sources of HONO), of HO2 (because of HO2 uptake on aerosol), and of NO3 (because of the equilibrium with N2O5). I don't think that a complete analysis of radical chemistry can just ignore these processes. If the authors think that heterogeneous chemistry is negligible under the conditions of this study (and it may well be so), they should provide some evidence or reason why that is the case.

The other important issue is the HONO/NO2 ratio which is set here to 2%, based on the Tan et al., 2019 paper. However that study examined Chinese megacities and I would expect HONO and NO2 levels to be different in rural areas. I appreciate that without HONO measurements it is not possible to be very accurate on this point, but since the paper shows that HONO is a major source of OH, this issue should be discussed somewhere in the manuscript. I suggest at least a sensitivity study to assess how the estimate of HONO impacts the model results and hence the conclusions of the paper. If, on the other hand, the conditions in this study and in the Tan et al., 2019, paper are similar, that would bring into question the classification of the measurement site as "rural", which would necessarily reframe the subject and the conclusions of the paper.

With regards to the analysis of ozone and formaldehyde I am confused about the model setup. On line 141 it is said that O3, NOx and VOCs (does it include HCHO?) are constrained in the model. However, on lines 147-151 and in Figure S1, "simulated ozone" is discussed. It is also not clear if HCHO is constrained or not. A species can be either constrained or calculated (simulated) in a model, but not both. The larger point, however, is that if O3 and/or HCHO are constrained, then the results in sections

3.4, 3.5, 3.6.2 and 3.6.3 need to be revised. It does not make much sense to look at the rate of production of a constrained variable because its value is set by the model and not calculated based on the values of the other variables. So the authors should first clarify whether O3 and HCHO are constrained or calculated in the model and then amend the discussion in sections 3.4, 3.5, 3.6.2 and 3.6.3 accordingly.

MINOR COMMENTS

In Table 1, and in the related text, I believe "42i" and "43i" need to be exchanged. Also, I suggest the detection limits and/or uncertainties are added to the Table 1.

line 103: correct to "Vaisala"

line 115: I imagine you mean 15m above the 5th floor?

line 142: correct to "RH"

lines 143: correct to "nitrous acid".

line 149: please define these indices (IOA, MB, NMB).

lines 199-201: I assume you are talking about simulated NO3 here. Please always make clear in the text, figures and captions, when you are talking about measurements and when about model results.

lines 205-207: I am not aware of RO+NO3 reactions forming RO2. Can you please clarify and/or correct?

line 212: correct to "ozonolysis".

---

## Referee Comment (RC2) · Anonymous Referee #2 · 23 Oct 2020

Summary: Air quality has become a serious issue in China. This study collected ambient concentration of several pollutants (e.g., O3 and HCHO) as well as precursors (NOx and VOCs) in a rural site of YRD in summer 2018 and applied an observation-based model (OBM) with MCM to investigate the impact of isoprene emissions on two oxidants: ozone and formaldehyde for five selected days. The way of evaluating the potential role of isoprene is based on changes of several simulated ROx radicals by removing isoprene from the 5-day baseline modeling from the baseline. It concludes that isoprene plays an important role in formation of ozone and formaldehyde since the

reduction of ROx radicals is significant when isoprene is removed.

The manuscript is reasonable written and consists of details needed to support the analysis and conclusions. However, several key questions need to be addressed before publication, as follows.

1) The title of manuscript "Observations and explicit modeling of isoprene chemical processing in polluted air masses in rural areas of the Yangtze River Delta region: radical cycling and formation of ozone and formaldehyde" is not well supported by the work presented. The observations are limited since key product species (i.e., MACR and MVK) and ROx radicals of isoprene were not measured or observed.

2) Is DSH a rural site? It is characterized as suburban by Lin et al. (2020) and impacted by a nearby freeway. Lin et al. (2020) indicate that both DSH and PD (urban site) are dominated by vehicle emissions sites (Figure 10) and isoprene emission is less in DSH than PD (Figure 5)? Can analysis be done for these five episodes in this study to demonstrate isoprene dominates among VOCs? Otherwise, it is hard to justify the study objective.

3) Model performance (i.e., OBM in this study) should be conducted against observed key species such as ozone, formaldehyde, and NOx before the model can be confidently used to simulate other key ROx species such as OH, HO2, RO, and RO2 (e.g., Figures 4, 5, 7-8). For instance, simulated local O3 is shown in Figure 7(A) but correlative discussion with observed O3 profile is needed. Similarly, simulated HCHO concentration in Figure 8(A) should be correlated with observed HCHO concentration. Without solid performance evaluation, simulated ROx radicals are questionable although they are comparable to other literature values, as indicated in this study.

4) As mention above, ROx radicals and key products (i.e., MACR and MVK) photochemically produced by isoprene and other precursors were not measured so model performance couldn't be conducted against these species. Without the validation, this is hard to evaluate the simulated ROx radicals with confidence, as mentioned above.

In addition, over 50 VOCs were measured but they were not utilized in this study. As an example, some VOCs primarily react with OH radical so those VOCs can be used as surrogates to estimate concentration of OH radicals, which can then be compared to simulated OH radicals. For example, Lin et al (2020) used X/E to estimate OH. Another analysis of VOC data can be conducted to evaluate the relative importance of isoprene in total VOCs. Isoprene has to be a significant part of VOCs emissions in order to achieve the objective of this study, evaluating isoprene's importance in rural areas.

5) Measurements of VOCs are described in details (Lines 116-125) but VOC analysis is lacking. Additional analysis would be useful. For instance, several types of VOCs (e.g., alkenes and aromatics) contribute to OVOC, an important specie focused in this study (in Figures 6 and 9), so their relationship to OVOC can be evaluated, in addition to the VOC analyses suggested above.

Technical comments:

1) Table 1: SO2 is listed as one of the measured pollutants but it is not used in this study at all. Please remove it from the table. CO is not listed here but shown in Figure 2.

2) Figure 2: CO concentration is almost flat so indicates this site is less impacted by traffic-related emissions. This contradicts with Lin et al (2020)'s observation (Figure 3), where NOx concentrations show traffic related variation in DSH.

3) Section 3.3 (line 210+): there is no discussion or description of Figure 5(B).

4) Figure 8: Net HCHO rate is negative for several hours around noon. What does that mean? Some discussion is needed.

5) Figures 6 and 9: It seems the red lines indicate photolysis production of ROx radicals while blue lines destruction or sink of these radicals. What does the black line represent? Some description is needed.

Minor comments:

1) Line 184: should be "series", not "serious"

2) Lines 512 and 517: these two references seemk identical.

3) Term "loss" is used in Figure 5 and its associated text while "destruction" or "sink" in Figures 7 and 8 and their description. They probably meant the same thing but consistency is preferred.

4) Line 205: "by separate the formation of RO2" should be revised for clarity. Do you mean "by separation from the formation of RO2"?

5) Line 263-264, the last sentence should be "Primary ROx sources and sinks are in red and blue, respectively."

---

## Author Comment (AC2) · 2 Dec 2020

**Response to Referee #2**

Received and published: 23 October 2020

1) The title of manuscript "Observations and explicit modeling of isoprene chemical processing in polluted air masses in rural areas of the Yangtze River Delta region: radical cycling and formation of ozone and formaldehyde" is not well supported by the work presented. The observations are limited since key product species (i.e., MACR and MVK) and ROx radicals of isoprene were not measured or observed.

Response: We are grateful for the comment. We admit that key product species of isoprene and ROx radicals of isoprene were not measured during our observation. Therefore, we changed the title into "Explicit modeling of isoprene chemical processing in polluted air masses in suburban areas of the Yangtze River Delta region: radical cycling and formation of ozone and formaldehyde"

2) Is DSH a rural site? It is characterized as suburban by Lin et al. (2020) and impacted by a nearby freeway. Lin et al. (2020) indicate that both DSH and PD (urban site) are dominated by vehicle emissions sites (Figure 10) and isoprene emission is less in DSH than PD (Figure 5)? Can analysis be done for these five episodes in this study to demonstrate isoprene dominates among VOCs? Otherwise, it is hard to justify the study objective. Response: Thanks for the reviewers' helpful advice. According to former studies (e.g. Lin et al. (2020)), DSL site is a suburban site. We have corrected our description in the revised manuscript. Under stagnant conditions, typically during early morning, DSL could be affected by nearby vehicle emissions. Figure 5 of Lin et al. (2020) exhibited the relative incremental reactivity (RIR) value of O3 precursors. In their study, RIR of isoprene is lower at DSH than that at PD site, suggesting that more O3 could be produced at PD site when a same proportion of isoprene was increased. However, at DSH, the ratio of the RIR from isoprene to RIR from anthropogenic hydrocarbons (AHC) is higher than that at PD site, indicating that as DSL site, isoprene plays a more important role than entire AHC in the secondary formation of O3. This is consistent with our objective, and we aim to investigate the influence of isoprene chemistry on O3 formation at YRD region. To roughly estimate the influence of isoprene on atmospheric oxidation capability, we adopted the approach presented in Zhu et al. (2020) to calculate the OH reactivity. The results suggest that isoprene, accounting for  $\sim 19\%$  of the total kOH, is the most significant VOC specie in terms of  $k_{OH}$ , with an average value of 0.89  $\pm$  0.44 s-1. This illustrates the significant role of isoprene in the photochemistry in suburban area. These descriptions have been added in the revised manuscript (Line 193-198).

Figure 5 Average relative incremental reactivity (RIR) values of the O3 precursors NO*x*, anthropogenic hydrocarbons (AHCs), biogenic hydrocarbons (NHCs), and CO at (a) the DSH site and (b) the PD site from 11–26 July.(from Lin et al. (2020))

3) Model performance (i.e., OBM in this study) should be conducted against observed key species such as ozone, formaldehyde, and NO*x* before the model can be confidently used to simulate other key RO*x* species such as OH, HO2, RO, and RO2 (e.g., Figures 4, 5, 7-8). For instance, simulated local O3 is shown in Figure 7(A) but correlative discussion with observed O3 profile is needed. Similarly, simulated HCHO concentration in Figure 8(A) should be correlated with observed HCHO concentration. Without solid performance evaluation, simulated RO*x* radicals are questionable although they are comparable to other literature values, as indicated in this study.

Response: We agree that solid performance evaluation is essential. Therefore, comparison of simulated and observed  $O_3$  and HCHO concentration are given in the supporting information. The discussion of simulated and observed  $O_3$  and HCHO concentrations are given in Line 153 to Line 164:

"The index of agreement (IOA), mean bias (MB) and normalized mean bias (NMB) are frequently used for model performance evaluation. These three parameters are calculated by Equation (2) to (4), where Si,  $O_i$ , and  $\overline{O}$  are the simulated, observed, and average value of the target compound. In this study, the IOA, MB and NMB of  $O_3$  was 0.90, 0.76 and 10%, respectively, suggesting that the model can reasonably reproduce the variations of  $O_3$ and could be used for further analysis. As for HCHO, the IOA, MB, and NMB was 0.74, 2.43, and 48%, respectively. In general, the model overestimated HCHO concentration, especially on July 29 and July 30. According to previous studies, the inconsistency between simulated and observed HCHO could be caused by the uncertainties in the treatment of dry deposition, faster vertical transport, uptake of HCHO, and fresh emission of precursor VOCs (Li et al., 2014). Nevertheless, these results still provide valuable information of secondary formation of HCHO at suburban area.".

The NO*x* concentrations are constrained by the observed value in our setup, so there is no need to compare the simulated and observed NO*x* concentration.

Reference:

Li, X., Rohrer, F., Brauers, T., Hofzumahaus, A., Lu, K., Shao, M., Zhang, Y. H., and Wahner, A.: Modeling of HCHO and CHOCHO at a semi-rural site in southern China during the PRIDE-PRD2006 campaign, Atmospheric Chemistry and Physics, 14, 12291-12305, 10.5194/acp-14-12291-2014, 2014.

4) As mention above, ROx radicals and key products (i.e., MACR and MVK) photochemically produced by isoprene and other precursors were not measured so model performance couldn't be conducted against these species. Without the validation, this is hard to evaluate the simulated ROx radicals with confidence, as mentioned above. In addition, over 50 VOCs were measured but they were not utilized in this study. As an example, some VOCs primarily react with OH radical so those VOCs can be used as surrogates to estimate concentration of OH radicals, which can then be compared to simulated OH radicals. For example, Lin et al (2020) used X/E to estimate OH. Another analysis of VOC data can be conducted to evaluate the relative importance of isoprene in total VOCs. Isoprene has to be a significant part of VOCs emissions in order to achieve the objective of this study, evaluating isoprene's importance in rural areas.

Response: Thank for the helpful advice. The average regional OH concentration (8.39  $\pm$  5.11 ×106 molecules cm-3) was estimated by E/X ratio, and relevant descriptions have been added in the revised manuscript. Please refer to Page 10, Line 228-233:

"To verify the performance of OBM model, regional mixing ratios of OH during daytime were also calculated by a parameterization method using ratios of measured ethylbenzene and m,p-xylene concentrations (see Text S2). The calculated average regional concentrations of OH ( $8.39 \pm 5.11 \times 10^6$  molecules cm-3) was in the same order of the magnitude of the OBM-simulated result ( $4.59 \pm 5.11 \times 10^6$  molecules cm-3), suggesting that the OBM-simulated radical concentration is reliable."

In addition, the OH reactivity ( $k_{OH}$ ) from VOCs was calculated and the result shows that isoprene alone can accounted for 19% of the total  $k_{OH}$ , indicating the significant role of isoprene in suburban area. Please refer to Page 10, Line 194-199:

"To roughly estimate the influence of isoprene on atmospheric oxidation capability, we adopted the approach given in the study of Zhu et al. (2020a) to calculate the OH reactivity ( $k_{OH}$ ). The result suggested that isoprene, accounting for ~19% of the total  $k_{OH}$ , was the most significant VOC specie from the perspective of  $k_{OH}$ , with an average value of 0.89 ± 0.44 s-1. This indicates the significant role of isoprene in the photochemistry in suburban area."

**Reference:**

Zhu, J., Cheng, H., Peng, J., Zeng, P., Wang, Z., Lyu, X., and Guo, H.: O3 photochemistry on O3 episode days and non-O3 episode days in Wuhan, Central China, Atmospheric Environment, 223, 10.1016/j.atmosenv.2019.117236, 2020a 5) Measurements of VOCs are described in detail (Lines 116-125) but VOC analysis is lacking. Additional analysis would be useful. For instance, several types of VOCs (e.g., alkenes and aromatics) contribute to OVOC, an important specie focused in this study (in Figures 6 and 9), so their relationship to OVOC can be evaluated, in addition to the VOC analyses suggested above.

Response: Thanks for the reviewer's helpful advice. Analysis about VOCs and the relationship between VOCs and OVOCs are added in the revised manuscript. Please refer to Line 194-214:

" To roughly estimate the influence of isoprene on atmospheric oxidation capability, we adopted the approach presented in Zhu et al. (2020) to calculate the OH reactivity ( $k_{OH}$ ). The results suggest that isoprene, accounting for ~19% of the total  $k_{OH}$ , is the most significant VOC species in terms of  $k_{OH}$ , with an average value of 0.89 ± 0.44 s-1. This illustrates the significant role of isoprene in the photochemistry in suburban area. Based on the explicit calculation, the total concentration of OVOC was obtained. Due to the complexity of OVOC formation, which involves hundreds of precursors for just one OVOC species, and the complicated chain reactions converting VOCs to OVOCs, it is difficult to give the accurate relationship between VOCs to OVOCs. Since VOCs were mainly oxidized by OH and O3 during daytime, we applied multi-linear regression model (given in Eq.(5) ) to provide the roughly relationship between VOCs and simulated OVOCs.

$$[OVOC] = \beta_0 + \beta_1[Alkane] + \beta_2[Alkene] + \beta_3[Aromatic] + \beta_4[OH]$$
(1)
+ \beta\_5[O\_3]

where  $\beta_0$ ,  $\beta_1$ ,  $\beta_2$ ,  $\beta_3$ ,  $\beta_4$ , and  $\beta_5$  are the coefficient from linear regression, [OVOC] and [OH] are the simulated concentration of OVOC and OH, respectively; [Alkane], [Alkene],

[Aromatic], [O3] are the observed concentration of alkane, alkene, aromatic, and O3. The Sig value and statistical reliability criteria (R) was 0.000 and 0.853 (shown in Table S3), respectively, indicating that the linear relationship represented by the equations (5) is statistically reliable. The  $\beta_1$ ,  $\beta_2$ ,  $\beta_3$  was 0.027, 0.623, and 0.820, respectively, suggesting that alkenes and aromatics are significant for the simulated OVOC concentration."

**Technical comments:**

Table 1: SO2 is listed as one of the measured pollutants but it is not used in this study at all. Please remove it from the table. CO is not listed here but shown in Figure 2.
 Response: We are grateful for this comment. The description of SO2 monitoring has been removed from table 1 and CO monitor has been added in table 1.

2) Figure 2: CO concentration is almost flat so indicates this site is less impacted by trafficrelated emissions. This contradicts with Lin et al (2020)'s observation (Figure 3), where NO*x* concentrations show traffic related variation in DSH.

Response: Thanks for the reviewer's careful comment. We agree that the CO concentration was almost flat during the scenarios, but when we look at the variation of NO, clearly peaks of NO was found during early morning, which is closed relate to traffic emission, and this result is also consistent with the observation of Lin et al. (2020), which also found clearly morning peak in DSL.

3) Section 3.3 (line 210+): there is no discussion or description of Figure 5(B).

Response: Thanks for the suggestion. Description of Figure 5 has been added in Line 386-390:"To investigate the underlying causes, we calculated the production rate of RO*x* (P(ROx)) and loss rate of RO*x* (L(ROx)) in S1, respectively (Figure 5 (B)). From the comparison, we found most of the reaction rates in P(ROx) and L(ROx) showed a decrease trend in S1, suggesting that the absence of isoprene slows down the RO*x* recycling."

4) Figure 8: Net HCHO rate is negative for several hours around noon. What does that mean? Some discussion is needed.

Response: The negative net HCHO rate around noon means the net reduction of HCHO. Relative description is given in Line 364-369:"Between 13:00 and 14:00, a negative net(HCHO) was found. Although the reaction of  $RO+O_2$  quickly produced HCHO at afternoon, the depletion pathways, especially the photolysis of HCHO, became more competitive, leading to the net reduction of HCHO. This also indicated that strong photochemical reactions do not monotonously profit the accumulation of HCHO, it can also constrain high HCHO levels in certain situations."

5) Figures 6 and 9: It seems the red lines indicate photolysis production of ROx radicals while blue lines destruction or sink of these radicals. What does the black line represent? Some description is needed.

Response: Thanks for this good suggestion. The black lines represent the processes in ROx recycling, and NOx recycling, and relative descriptions has been added for Figure 6 and Figure 9.

Minor comments:

Line 184: should be "series", not "serious"
 Response: We have recorrected this mistake as suggested.

2) Lines 512 and 517: these two references seem identical.

Response: We have removed the replicated reference.

3) Term "loss" is used in Figure 5 and its associated text while "destruction" or "sink" in Figures 7 and 8 and their description. They probably meant the same thing but consistency is preferred.

Response: Thanks for the helpful suggestion, the term "loss", "destruction" and "sink" has been unified in our manuscript.

4) Line 205: "by separate the formation of RO2" should be revised for clarity. Do you mean"by separation from the formation of RO2"?

Response: Thanks for the reviewer's suggestion, we have changed this sentence into "By separating the formation pathways of RO2".

5) Line 263-264, the last sentence should be "Primary ROx sources and sinks are in red and blue, respectively."

Response: We have revised this sentence as suggested.

[revised manuscript text omitted]

| CO                                                                  | Model 48i, Thermo
Fischer Scientific, USA | <mark>60 s</mark> | <mark>40 ppbv</mark>                   |
| НСНО                                                                | AL4021, Aero-Laser, GER                      | 90 s              | 0.1 ppbv                               |
| VOCs species                                                        | GC866, Agilent., USA                         | 1 hour            | -                                      |
| Temperature, relative
humidity, wind speed and
wind direction | Meteorological station,
Vaisala, NLD      | 60 s              | •                                      |

The measuring instruments are shown in Table 1. Wind speed (WS), wind direction (WD), temperature (T), and relative humidity (RH) were simultaneously observed by a meteorological station (Vaisala., FIN). According to China's air quality standard, several criteria air pollutants were measured during this experiment. O3 was measured by an ultraviolet photometric analyzer (Model 49i, Thermo Fischer Scientific., USA), which has a detection limit of 0.5 ppbv at 60 second resolution. 1 min resolution of nitrogen oxides (NO and NO2) data were simultaneously observed by a chemiluminescence instrument (Model 42i, Thermo Fischer Scientific., USA), which has a detection limit of 0.4 ppbv. Carbon monoxide was monitored by a gas filter correlation infrared absorption analyzer (Model 48i, Thermo Fischer Scientific., USA), which has a detection limit of 0.04 ppm. All the online instruments used for gas analyzer were auto-zero every day, and were multipoint calibrated every month. All the instruments used for the online observation were housed on top of a 5-floor-high building, which was about 15 m above the ground level.

A total of 55 VOC species, including 28 alkanes, 10 alkenes, 16 aromatics and acetylene were continuously analyzed at our sampling site by two online gas chromatograph with flame ionization detector (GC-FID) systems (GC-866 airmoVOC C2- $C_6$  #58850712 and airmoVOC C6- $C_{12}$  #283607112, Agilent., USA) with a time resolution of 1 hour during our experiment. Ambient samples are directly inhaled into this system by a pump. Low carbon VOCs (C2-C6) are captured by a low temperature (-10 °C) preconcentration system, while high carbon VOCs are concentrated by a built-in room temperature preconcentration system. Then the preconcentration systems are heated and desorb VOCs, which are then carried into chromatographic columns by helium. Individual

VOCs separated in the columns are eventually detected by FID systems. Formaldehyde (HCHO) was continuously measured by a Hantzsch fluorescence technique (AL4201, Aerolaser GmbH., GER), which is based on fluorometric Hantzsch reaction in the liquid phase, requiring the quantitative transfer of HCHO from gas phase to liquid phase. A Hantzsch reagent (acetylacetone) was used in this instrument.

**2.2 Observation-based model**

A user-friendly zero-dimensional (0-D) box model (F0AM) was used to simulate the chemical processes in the atmosphere in this study. This model was developed by Wolfe et al.(2016b) based on University of Washington Chemical Model (UWCM). Dry deposition, aloft exchange, and atmospheric dilution were considered in this model. We chose the Master Chemical Mechanism (MCM) v3.3.1 as the chemical mechanism with more than 5,800 chemical species and 17,000 reactions, which enables a detailed description of the complex reactions. In addition to gas-phase reactions, several heterogenous processes including the uptake of HO2, N2O5 and HCHO on aerosol surface, and heterogenous source of HONO were also considered in our simulation. These reactions rate constants and uptake coefficient were obtained from the study of Riedel et al. (2014), Xue et al. (2014) and Li et al. (2014). Since key parameters such as aerosol surface areas (SA) and particle diameters (r) were not measure during our observation, an average SA (640 nm2/cm3) was obtained from the field campaign in Shanghai (Wang et al., (2014)).

| Reactions                          | Reaction rate constant                                               | Reference        |
|------------------------------------|----------------------------------------------------------------------|-------------------------|
|                                    | $\gamma \omega S_A/4$ (for CLNO 2 formation)              | Riedel et               |
| $N_2O_5 \rightarrow CLNO_2 + HNO3$ | $(2 - \emptyset)\gamma\omega S_A/4$ (for HNO 3 formation) | <mark>al. (2014)</mark> |

---

## Author Response (AR1)

**Response to Referee #1**

Major comments

I find two main issues with the analysis presented in sections 3.2 and 3.3. One is the lack of heterogeneous chemistry in the model. This is likely to impact both the levels and the budget of OH (because of the heterogeneous sources of HONO), of $HO_2$ (because of $HO_2$ uptake on aerosol), and of $NO_3$ (because of the equilibrium with $N_2O_5$). I don't think that a complete analysis of radical chemistry can just ignore these processes. If the authors think that heterogeneous chemistry is negligible under the conditions of this study (and it may well be so), they should provide some evidence or reason why that is the case.

Response: We really appreciate the reviewers careful and valuable comments. We agree that the heterogeneous processes associated with HONO source, uptake of $HO_2$ and $N_2O_5$ could be important for the budget of OH, $HO_2$ and $NO_3$ under certain conditions. Therefore, we tested the heterogeneous processes including uptake of $N_2O_5$, HCHO, and $HO_2$ on aerosols surface, and heterogenous sources of HONO in our simulation, as summarized in Table 1. Rate constants and uptake coefficients for these reactions were obtained from previous studies (Riedel et al. (2014); Xue et al. (2014); Li et al. (2014)). Since key parameters such as aerosol surface areas ($S_A$) were not directly measured during our observation period, an average value of $S_A$ (640 $nm^2/cm^3$ from the study of Wang et al. (2014)) was adopted in this study. Our results suggest that adding heterogenous processes in our simulation could lead to decrease of OH, $HO_2$, $RO_2$ and $NO_3$ by 1.53%, 4.54%, 2.73%

and 6.53%, respectively. These processes have been included in our base simulation and results are updated accordingly.

Table 1. Additional heterogenous reactions and associated rate constants used by the model

| Reactions | Rate constants | No. |
|---|---|---|
| $N_2O_5 \rightarrow CLNO_2 + HNO3$ | $\gamma\omega S_A/4$(for CLNO$_2$ formation)
$(2-\varnothing)\gamma\omega S_A/4$(for HNO$_3$ formation) | Riedel et al. (2014) |
| $NO_2 \rightarrow HONO$ | $k_g = \dfrac{1}{8} \times \omega\gamma(\dfrac{S}{V})$
$k_a = \dfrac{1}{4}\omega\gamma S_A$ | Xue et al. (2014) |
| $HO_2 \rightarrow products$ | $k = (\dfrac{r}{D_g} + \dfrac{4}{\gamma}\omega)^{-1}S_A$ | Xue et al. (2014) |
| $HCHO \rightarrow products1$ | $k = \dfrac{1}{4}\omega\gamma S_A$ | Li et al. (2014) |

$\gamma$= uptake coefficient for the given reactant with aerosol surface area; $\phi$ = product yield; $\omega$=mean molecular speed of the given reactant (m/s); $S_A$=RH corrected aerosol surface area concentration (nm$^2$/cm$^3$); r=surface-weighted particle radius.

References:

Li, X., Rohrer, F., Brauers, T., Hofzumahaus, A., Lu, K., Shao, M., Zhang, Y. H., and Wahner, A. (2014). Modeling of HCHO and CHOCHO at a semi-rural site in southern China during the PRIDE-PRD2006 campaign, Atmospheric Chemistry and Physics, 14, 12291-12305, 10.5194/acp-14-12291-2014.

Riedel, T. P., Wolfe, G. M., Danas, K. T., Gilman, J. B., Kuster, W. C., Bon, D. M., Vlasenko, A., Li, S.-M., Williams, E. J., Lerner, B. M., Veres, P. R., Roberts, J. M., Holloway, J. S., Lefer, B., Brown, S. S., and Thornton, J. A. (2014). An MCM modeling study of nitryl chloride (ClNO2) impacts on oxidation, ozone production and nitrogen oxide partitioning in polluted continental outflow, Atmos. Chem. Phys., 14, 3789–3800, https://doi.org/10.5194/acp-14-3789-2014.

Xue, L., Wang, T., Gao, J., Ding, A., Zhou, X., Blake, D. R., Fang, X., Saunders, S. M., Fan, S., Zuo, H., Zhang, Q., Wang, W. (2014). Ground-level ozone in four Chinese cities: precursors, regional transport and heterogeneous processes. Atmospheric chemistry and physics, 14(23), 13175-13188.

Wang, X., Chen, J., Cheng, T., Zhang, R., Wang, X. (2014) Particle number concentration, size distribution and chemical composition during haze and photochemical smog episodes in Shanghai[J]. Journal of Environmental Sciences, 2014, 26(009):1894-1902.

The other important issue is the HONO/NO$_2$ ratio which is set here to 2%, based on the Tan et al., 2019 paper. However that study examined Chinese megacities and I would expect HONO and NO$_2$ levels to be different in rural areas. I appreciate that without HONO measurements it is not possible to be very accurate on this point, but since the paper shows that HONO is a major source of OH, this issue should be discussed somewhere in the manuscript. I suggest at least a sensitivity study to assess how the estimate of HONO impacts the model results and hence the conclusions of the paper. If, on the other hand, the conditions in this study and in the Tan et al., 2019, paper are similar, that would bring into question the classification of the measurement site as "rural", which would necessarily reframe the subject and the conclusions of the paper.

Response: We agree that the HONO/NO$_2$ ratio in this study is different from that in Tan et al. (2019). To investigate the sensitivity of our results to the HONO/NO$_2$ ratio, a series of simulations with different HONO/NO$_2$ ratios were conducted and the results were summarized in Table 2. A lower HONO/NO$_2$ ratio (e.g. 0.005) could lead to decrease of OH radical by 15.3% and a higher ratio (e.g. 0.04) could increase OH concentration by

14.1%. This could be explained by the important role of HONO photolysis as one of the OH sources. Discussions on the sensitivity results have been added to the revised manuscript (Page 13, Line 258-266):

"Sensitive studies were conducted to quantify the influences of different $HONO/NO_2$ ratios on radical recycling (Text S3, Figure. S1 and Table S1). As expected, lower $HONO/NO_2$ ratio leads to lower HONO concentrations, and subsequent less OH generated from the photolysis of HONO. The sensitive study shows that when $HONO/NO_2$ ratio is 0.005, the daytime OH level could decrease by 15.3%. Vice versa, a higher $HONO/NO_2$ ratio (e.g., 0.04) can promote OH concentration by 14.08%. This result indicates that the photolysis of HONO is essential to the generation of OH, and therefore a simultaneous measurement of HONO is highly recommended for analyzing local radical recycling."

[Figure]

Fig. 1 Comparison of OH concentration under different $HONO/NO_2$ ratios.

**Table 2 Model sensitivity test result.**

| HONO/NO$_2$ ratio | Change in OH (%) |
| --- | --- |
| 0.005 | -15.3% |
| 0.01 | -9.3% |
| 0.03 | 7.5% |
| 0.04 | 14.1% |

With regards to the analysis of ozone and formaldehyde I am confused about the model setup. On line 141 it is said that O$_3$, NO$x$ and VOCs (does it include HCHO?) are constrained in the model. However, on lines 147-151 and in Figure S1, "simulated ozone" is discussed. It is also not clear if HCHO is constrained or not. A species can be either constrained or calculated (simulated) in a model, but not both. The larger point, however, is that if O$_3$ and/or HCHO are constrained, then the results in sections 3.4, 3.5, 3.6.2 and 3.6.3 need to be revised. It does not make much sense to look at the rate of production of a constrained variable because its value is set by the model and not calculated based on the values of the other variables. So the authors should first clarify whether O$_3$ and HCHO are constrained or calculated in the model and then amend the discussion in sections 3.4, 3.5, 3.6.2 and 3.6.3 accordingly.

Response: O$_3$ and HCHO were not constrained in our simulation, because we want to analyze the secondary formation of these compounds. To avoid misunderstanding, we have revised related descriptions in the revised manuscript (Line 146-147): " Hourly averaged concentrations of speciated VOCs (except HCHO), NO, NO$_2$ and meteorological parameters (such as T, RTH, P) were used to constrain the F0AM model."

Minor comments

In Table 1, and in the related text, I believe "42i" and "43i" need to be exchanged. Also, I suggest the detection limits and/or uncertainties are added to the Table 1.

Response: Thanks for the helpful advice. We have exchanged "42i" and "43i" in Table 1 and in the related text and add detection limits in Table 1.

line 103: correct to "Vaisala"

Response: We have corrected "Visala" into "Vaisala"

line 115: I imagine you mean 15m above the 5th floor?

Response: We are sorry for the unclear expression, and we mean the top of the 5-floor-high building is 15 m above the ground level. Therefore, we have revised this sentence into "The instruments were housed on the top of a 5-floor-high building, which was about 15 m above the ground level."

line 142: correct to "RH".

Response: We have revised this word as suggested.

lines 143: correct to "nitrous acid".

Response: We have revised this word as suggested.

line 149: please define these indices (IOA, MB, NMB).

Response: The definition of IOA, MB and NMB are given in Line 153-157: "The index of agreement (IOA) , mean bias (MB) and normalized mean bias (NMB) are frequently used to estimate the model performance. These three parameters can be calculated by Equation (2) to (4), where $S_i$, $O_i$, and $\bar{O}$ are the simulated, observed, and average value of the target compound. "

$$IOA = 1 - \frac{\sum(S_i - O_i)^2}{\sum(|S_i - \bar{O}| + |O_i - \bar{O}|)^2} \tag{2}$$

$$MB = \frac{\sum(S_i - O_i)}{N} \tag{3}$$

$$NMB = \frac{\sum(S_i - O_i)}{\sum O_i} \times 100 \tag{4}$$

lines 199-201: I assume you are talking about simulated $NO_3$ here. Please always make clear in the text, figures and captions, when you are talking about measurements and when about model results.

Response: Thanks for the reviewer's valuable suggestion. We have specified the "simulated $NO_3$" in Line 236-237. In addition, we have checked the similar unclear expression throughout the paper.

lines 205-207: I am not aware of $RO+NO_3$ reactions forming $RO_2$. Can you please clarify and/or correct?

Response: We are sorry for this mistake. It should be the reaction of $VOCs+NO_3$ that account for over 70% $RO_2$ production during nighttime, and relevant description has been revised.

line 212: correct to "ozonolysis".

Response: We have corrected this error as suggested.

**Response to Referee #2**

1) The title of manuscript "Observations and explicit modeling of isoprene chemical processing in polluted air masses in rural areas of the Yangtze River Delta region: radical cycling and formation of ozone and formaldehyde" is not well supported by the work presented. The observations are limited since key product species (i.e., MACR and MVK) and $RO_x$ radicals of isoprene were not measured or observed.

Response: We are grateful for the comment. We admit that key product species of isoprene and $RO_x$ radicals of isoprene were not measured during our observation. Therefore, we changed the title into "Explicit modeling of isoprene chemical processing in polluted air masses in suburban areas of the Yangtze River Delta region: radical cycling and formation of ozone and formaldehyde"

2) Is DSH a rural site? It is characterized as suburban by Lin et al. (2020) and impacted by a nearby freeway. Lin et al. (2020) indicate that both DSH and PD (urban site) are dominated by vehicle emissions sites (Figure 10) and isoprene emission is less in DSH than PD (Figure 5)? Can analysis be done for these five episodes in this study to demonstrate isoprene dominates among VOCs? Otherwise, it is hard to justify the study objective.

Response: Thanks for the reviewers' helpful advice. According to former studies (e.g. Lin et al. (2020)), DSL site is a suburban site. We have corrected our description in the revised manuscript. Under stagnant conditions, typically during early morning, DSL could be affected by nearby vehicle emissions. Figure 5 of Lin et al. (2020) exhibited the relative incremental reactivity (RIR) value of $O_3$ precursors. In their study, RIR of isoprene is lower at DSH than that at PD site, suggesting that more $O_3$ could be produced at PD site when a same proportion of isoprene was increased. However, at DSH, the ratio of the RIR from isoprene to RIR from anthropogenic hydrocarbons (AHC) is higher than that at PD site, indicating that as DSL site, isoprene plays a more important role than entire AHC in the secondary formation of $O_3$. This is consistent with our objective, and we aim to investigate the influence of isoprene chemistry on $O_3$ formation at YRD region. To roughly estimate the influence of isoprene on atmospheric oxidation capability, we adopted the approach presented in Zhu et al. (2020) to calculate the

OH reactivity. The results suggest that isoprene, accounting for ~19% of the total $k_{OH}$, is the most significant VOC specie in terms of $k_{OH}$, with an average value of $0.89 \pm 0.44$ $s^{-1}$. This illustrates the significant role of isoprene in the photochemistry in suburban area. These descriptions have been added in the revised manuscript (Line 193-198).

[Figure]

Figure 5 Average relative incremental reactivity (RIR) values of the $O_3$ precursors NO$x$, anthropogenic hydrocarbons (AHCs), biogenic hydrocarbons (NHCs), and CO at (a) the DSH site and (b) the PD site from 11–26 July.(from Lin et al. (2020))

3) Model performance (i.e., OBM in this study) should be conducted against observed key species such as ozone, formaldehyde, and NO$x$ before the model can be confidently used to simulate other key RO$x$ species such as OH, $HO_2$, RO, and $RO_2$ (e.g., Figures 4, 5, 7-8). For instance, simulated local $O_3$ is shown in Figure 7(A) but correlative discussion with observed

$O_3$ profile is needed. Similarly, simulated HCHO concentration in Figure 8(A) should be correlated with observed HCHO concentration. Without solid performance evaluation, simulated RO$x$ radicals are questionable although they are comparable to other literature values, as indicated in this study.

Response: We agree that solid performance evaluation is essential. Therefore, comparison of simulated and observed $O_3$ and HCHO concentration are given in the supporting information. The discussion of simulated and observed $O_3$ and HCHO concentrations are given in Line 153 to Line 164:

"The index of agreement (IOA), mean bias (MB) and normalized mean bias (NMB) are frequently used for model performance evaluation. These three parameters are calculated by Equation (2) to (4), where $S_i$, $O_i$, and $\bar{O}$ are the simulated, observed, and average value of the target compound. In this study, the IOA, MB and NMB of $O_3$ was 0.90, 0.76 and 10%, respectively, suggesting that the model can reasonably reproduce the variations of $O_3$ and could be used for further analysis. As for HCHO, the IOA, MB, and NMB was 0.74, 2.43, and 48%, respectively. In general, the model overestimated HCHO concentration, especially on July 29 and July 30. According to previous studies, the inconsistency between simulated and observed HCHO could be caused by the uncertainties in the treatment of dry deposition, faster vertical transport, uptake of HCHO, and fresh emission of precursor VOCs (Li et al., 2014). Nevertheless, these results still provide valuable information of secondary formation of HCHO at suburban area.".

The NO$x$ concentrations are constrained by the observed value in our setup, so there is no need to compare the simulated and observed NO$x$ concentration.

to estimate OH. Another analysis of VOC data can be conducted to evaluate the relative importance of isoprene in total VOCs. Isoprene has to be a significant part of VOCs emissions in order to achieve the objective of this study, evaluating isoprene's importance in rural areas.

Response: Thank for the helpful advice. The average regional OH concentration (8.39 ± 5.11

$\times10^6$ molecules cm$^{-3}$) was estimated by E/X ratio, and relevant descriptions have been added in the revised manuscript. Please refer to Page 10, Line 228-233:

"To verify the performance of OBM model, regional mixing ratios of OH during daytime were also calculated by a parameterization method using ratios of measured ethylbenzene and m,p- xylene concentrations (see Text S2). The calculated average regional concentrations of OH

(8.39 ± 5.11 $\times10^6$ molecules cm$^{-3}$) was in the same order of the magnitude of the OBM- simulated result (4.59 ± 5.11 $\times10^6$ molecules cm$^{-3}$), suggesting that the OBM-simulated radical concentration is reliable."

In addition, the OH reactivity (k$_{OH}$) from VOCs was calculated and the result shows that isoprene alone can accounted for 19% of the total k$_{OH}$, indicating the significant role of isoprene in suburban area. Please refer to Page 10, Line 194-199:

"To roughly estimate the influence of isoprene on atmospheric oxidation capability, we adopted the approach given in the study of Zhu et al. (2020a) to calculate the OH reactivity (k$_{OH}$). The result suggested that isoprene, accounting for ~19% of the total k$_{OH}$, was the most significant

VOC specie from the perspective of $k_{OH}$, with an average value of $0.89 \pm 0.44$ s$^{-1}$. This indicates the significant role of isoprene in the photochemistry in suburban area."

species in terms of $k_{OH}$, with an average value of $0.89 \pm 0.44$ s$^{-1}$. This illustrates the significant role of isoprene in the photochemistry in suburban area. Based on the explicit calculation, the total concentration of OVOC was obtained. Due to the complexity of OVOC formation, which involves hundreds of precursors for just one OVOC species, and the complicated chain reactions converting VOCs to OVOCs, it is difficult to give the accurate relationship between

VOCs to OVOCs. Since VOCs were mainly oxidized by OH and $O_3$ during daytime, we applied multi-linear regression model (given in Eq.(5) ) to provide the roughly relationship between

VOCs and simulated OVOCs.

$$[OVOC] = \beta_0 + \beta_1[Alkane] + \beta_2[Alkene] + \beta_3[Aromatic] + \beta_4[OH] + \beta_5[O_3] \quad (1)$$

where $\beta_0$, $\beta_1$, $\beta_2$, $\beta_3$, $\beta_4$ , and $\beta_5$ are the coefficient from linear regression, [OVOC] and [OH]

are the simulated concentration of OVOC and OH, respectively; [Alkane], [Alkene],

[Aromatic], [$O_3$] are the observed concentration of alkane, alkene, aromatic, and $O_3$. The Sig value and statistical reliability criteria (R) was 0.000 and 0.853 (shown in Table S3), respectively, indicating that the linear relationship represented by the equations (5) is statistically reliable. The $\beta_1$, $\beta_2$, $\beta_3$ was 0.027, 0.623, and 0.820, respectively, suggesting that alkenes and aromatics are significant for the simulated OVOC concentration."

Technical comments:

1) Table 1: $SO_2$ is listed as one of the measured pollutants but it is not used in this study at all.

Please remove it from the table. CO is not listed here but shown in Figure 2.

Response: We are grateful for this comment. The description of $SO_2$ monitoring has been removed from table 1 and CO monitor has been added in table 1.

2) Figure 2: CO concentration is almost flat so indicates this site is less impacted by traffic- related emissions. This contradicts with Lin et al (2020)'s observation (Figure 3), where NO$x$

concentrations show traffic related variation in DSH.

Response: Thanks for the reviewer's careful comment. We agree that the CO concentration was almost flat during the scenarios, but when we look at the variation of NO, clearly peaks of

NO was found during early morning, which is closed relate to traffic emission, and this result is also consistent with the observation of Lin et al. (2020), which also found clearly morning peak in DSL.

3) Section 3.3 (line 210+): there is no discussion or description of Figure 5(B).

Response: Thanks for the suggestion. Description of Figure 5 has been added in Line 386-

390:"To investigate the underlying causes, we calculated the production rate of $RO_x$ ($P(RO_x)$)

and loss rate of $RO_x$ ($L(RO_x)$) in S1, respectively (Figure 5 (B)). From the comparison, we found most of the reaction rates in $P(RO_x)$ and $L(RO_x)$ showed a decrease trend in S1, suggesting that the absence of isoprene slows down the $RO_x$ recycling."

4) Figure 8: Net HCHO rate is negative for several hours around noon. What does that mean?

Some discussion is needed.

Response: The negative net HCHO rate around noon means the net reduction of HCHO.

Relative description is given in Line 364-369:"Between 13:00 and 14:00, a negative net(HCHO)

was found. Although the reaction of $RO+O_2$ quickly produced HCHO at afternoon, the depletion pathways, especially the photolysis of HCHO, became more competitive, leading to the net reduction of HCHO. This also indicated that strong photochemical reactions do not monotonously profit the accumulation of HCHO, it can also constrain high HCHO levels in certain situations."

5) Figures 6 and 9: It seems the red lines indicate photolysis production of $RO_x$ radicals while blue lines destruction or sink of these radicals. What does the black line represent? Some description is needed.

Response: Thanks for this good suggestion. The black lines represent the processes in $RO_x$

recycling, and $NO_x$ recycling, and relative descriptions has been added for Figure 6 and Figure

9.

Minor comments:

1) Line 184: should be "series", not "serious"

Response: We have recorrected this mistake as suggested.

2) Lines 512 and 517: these two references seem identical.

Response: We have removed the replicated reference.

3) Term "loss" is used in Figure 5 and its associated text while "destruction" or "sink" in

Figures 7 and 8 and their description. They probably meant the same thing but consistency is preferred.

Response: Thanks for the helpful suggestion, the term "loss", "destruction" and "sink" has been unified in our manuscript.

4) Line 205: "by separate the formation of $RO_2$" should be revised for clarity. Do you mean

"by separation from the formation of $RO_2$"?

Response: Thanks for the reviewer's suggestion, we have changed this sentence into "By separating the formation pathways of $RO_2$".

5) Line 263-264, the last sentence should be "Primary $RO_x$ sources and sinks are in red and blue, respectively."

Response: We have revised this sentence as suggested.

Revised manuscript

[revised manuscript text omitted]

---

## Author Response (AR2)

**Point-by-point response to comments by Reviewer#1**

We thank the reviewer for the detailed and constructive review comments. Below is our point-by-point response to each comment, marked in blue. Changes made to the main text are also marked in blue in the revised manuscript file.

**Major Comments:**

1. Many important parameters were not measured, including some that the authors themselves indicate as very important: for example, $CH_4$, HONO, photolysis rates, aerosol surface area. There is no attempt to estimate the uncertainty of the model results derived from these missing constraints. In addition, the instrument used to measure $NOx$ has known interferences, especially when it comes to $NO_2$. This can affect the results but it is not mentioned at all.

**Response**: Thanks for raising this issue. It is unfortunate that some important parameters were not measured. We'll detail below the rationality of this study based on sensitivity analysis and discuss associated uncertainties. We have also revised the manuscript text accordingly.

**1. Sensitivity analysis for those related parameters that have not been measured.**

We agree that it would be much better to involve online measurement of $CH_4$, HONO, photolysis rates, and aerosol surface area in this study. However, these measurements were not conducted due to the absence of those instruments. To reduce uncertainty, we conducted sensitivity analyses of modelled $O_3$, HCHO, and OH concentrations at different $CH_4$ concentrations, $HONO/NO_2$ ratio, photolysis rates, and aerosol surface area (Figure 1 to 4 below). Results show that when $HONO/NO_2$ ratio is 0.005, the daytime OH level could decrease by 15.28%. Vice versa, a higher $HONO/NO_2$ (e.g., 0.04) can promote OH concentration by 14.08%. This result indicates that the photolysis of HONO is essential for the generation of OH, and therefore a simultaneous measurement of HONO is highly recommended for the analysis of local radical recycling in the future. The sensitivity analyses show that the $O_3$, HCHO and OH concentrations could increase by 51.14%, 34.52%, and 50.38%, respectively, when photolysis rates were increased by 40%. On the contrary, when photolysis rates were decreased by 40%, $O_3$, HCHO and OH concentration could decrease by 50.59%, 30.84%, and 47.24%, respectively. However, the modelled $O_3$, HCHO, and OH

concentration did not show obvious changes when $CH_4$ concentrations and aerosol surface area (SA) changes.

**2. Uncertainty regarding NOx measurement**

We agree that the instrument (Model 42i, Thermo Fischer Scientific, USA) used to measure $NOx$ has interferences. This instrument is widely used in atmospheric research, and according to the study of Xu et al. (2013), this analyzer could accurately measure the NO concentration, while overestimate the concentration of $NO_2$ to some extent. However, the overestimation of $NO_2$ could not be precisely quantified without the observation of the actual $NO_2$ concentration (usually by photolytic converter). Here, to test the potential overestimates of $NO_2$ concentrations by our $NOx$ analyzer, we calculated the changes in $O_3$, OH, and HCHO concentrations when cutting $NO_2$ concentration by 10% ~ 40% (Figure 5). Results suggest that decreasing $NO_2$ could lead to increase or decrease of $O_3$, HCHO and OH concentrations in different cases. Overall, decreasing $NO_2$ by 40% could cause 6.94%, 12.07%, and 6.29% increase in $O_3$, HCHO, and OH concentrations, respectively. Therefore, more accurate observation of $NO_2$ should be considered in future studies. The following figures have been inserted to the supporting material.

**3. Uncertainties regarding model results**

According to the sensitivity analysis described above, when $HONO/NO_2$ ratio is 0.005, the daytime OH level could decrease by 15.28%. Vice versa, a higher $HONO/NO_2$ (e.g., 0.04) can promote OH concentration by 14.08%. However, the modelled $O_3$, HCHO, and OH concentration did not show obvious changes when $CH_4$ and aerosol surface area (SA) input changes. As for photolysis rate, the $O_3$, HCHO and OH concentration could increase by 51.14%, 34.52%, and 50.38%, respectively, when photolysis rates were increased by 40%. On the contrary, when photolysis rates were decreased by 40%, $O_3$, HCHO and OH concentration could decrease by 50.59%, 30.84%, and 47.24%, respectively. Furthermore, decreasing $NO_2$ by 40% could cause 6.94%, 12.07%, and 6.29% increase in $O_3$, HCHO, and OH concentrations, respectively.

[Figure]

**Figure 1.** Sensitivity analysis of OBM modelled $O_3$, HCHO, and OH concentrations with different $CH_4$ concentrations.

[Figure]

**Figure 2.** Comparison of OH concentration under different $HONO/NO_2$ ratios.

[Figure]

**Figure 3.** Sensitivity analysis of OBM modelled O$_3$, HCHO, and OH concentrations with different photolysis rates.

[Figure]

**Figure 4.** Sensitivity analysis of OBM modelled O$_3$, HCHO, and OH concentrations with different SA values

[Figure]

**Figure 5.** Sensitivity analysis of OBM modelled $O_3$, HCHO, and OH concentrations with reduced $NO_2$ concentrations.

**Reference**

Xu, Z., Wang, T., Xue, L. K., Louie, P. K. K., Luk, C. W. Y., Gao, J., Wang, S. L., Chai, F. H., and Wang, W. X.: Evaluating the uncertainties of thermal catalytic conversion in measuring atmospheric nitrogen dioxide at four differently polluted sites in China, Atmos. Environ., 76, 221–226, 2013.

**Relevant description has been inserted to the manuscript as follows, and the related figures have been inserted to the SI:**

Due to limitations in the observations, several issues should be noted in the application of the OBM model to evaluate the local chemistry in the present study. Firstly, methane concentration, which was set to 1850 ppbv based on previous observations, could be an overestimation or underestimation. Thus, we conducted sensitivity analysis of modelled $O_3$, OH, and HCHO with different methane values (from 1600 ppbv to 1900 ppbv) (Figure S7). The model predicted $O_3$, HCHO, and OH concentration with negligible change under different $CH_4$ values. Secondly, the photolysis rates directly influence the key photochemical processes during the day. Since the photolysis rates were not measured during the sampling period, we also conducted sensitivity analysis by increasing or decreasing the photolysis rates by 20% and 40%. Results showed that the $O_3$, HCHO and OH concentration could increase by 51.14%, 34.52%, and 50.38%, respectively, when photolysis rates were increased by 40% (Figure S8). On the contrary, when photolysis rates were decreased by 40%, $O_3$, HCHO and OH concentration

decreased by 50.59%, 30.84%, and 47.24%, respectively (Figure S6). According to the study by Xu et al. (2013), $NO_2$ concentration measured by the molybdenum oxide converter technique can be significantly overestimated in areas far away from fresh $NOx$ emission sources. Therefore, OBM simulations with reduced $NO_2$ concentrations were conducted. The results suggest that decreasing $NO_2$ could increase or decrease of $O_3$, HCHO and OH concentrations under different scenarios (Figure S9). Overall, decreasing $NO_2$ by 40% could cause 6.94%, 12.07%, and 6.29% increase in $O_3$, HCHO, and OH concentrations, respectively. Finally, the total surface area of aerosols was obtained from the study of Wang et al. (2014) and the uncertainty of this value could directly influence the heterogeneous reactions in this model. Therefore, we conducted sensitive analysis by using increasing or decreasing SA value by 40% (Figure S10). The results show that $O_3$, HCHO, and OH concentrations did not exhibit obvious changes when SA changed. Hence, accurate measurement data of photolysis rate and $NO_2$ concentration is strongly recommended in further OBM analyses.

2. $O_3$ and HCHO concentrations are affected by several non-chemical processes, which are difficult to account for with a zero-dimensional model. Although dilution, "aloft exchange" and deposition are mentioned at some point, they are not addressed in the discussion. It is legitimate to use such a model to focus on the in-situ photochemical pathways that form and destroy $O_3$ and HCHO, but then it should be made very clear that the analysis is limited to those processes and to local conditions.

**Response**: Thanks for the constructive comment. We agree that the zero-dimension OBM model indeed has limitations. For better clarification, we have addressed that the discussion of model results is limited to local condition in the revised manuscript.

**Revised manuscript (Page 10, Line 193-196):**
"To investigate the impact of local chemistry on ozone formation and avoid the influence of emission transportation, five days under stagnant condition (with daily average wind speed less than 2m/s and maximum daily 8-h average (MDA8) $O_3$ concentration >75 ppb) were identified as typical local chemistry cases."
Page 13, Line 251-253:

"It should be noted that, the discussion below is limited to local conditions (cases with average wind speed lower than 2m/s), since transportation of emissions are not considered in the 0-dimensional model."

Page 25, Line 503-505:

"Our observations at a suburban site of the YRD region from April to June in 2018 captured 5 typical local $O_3$ formation episodes. The detailed atmospheric photochemistry during these episodes were analyzed by a typical 0-D box model on a local scale."

3. It is hard to assess the reliability of the model results without radical measurements, and in the absence of a sensitivity study. As such, the radical budgets (Fig. 5 and 6) are qualitative at best. Previous studies may help with an estimate of the level of agreement between model and measurements, but only to the extent that the models being compared are similar. Moreover, there is some discussion of nocturnal processes, but the results of night-time chemistry are even more uncertain than those of day-time chemistry, because the model is not necessarily good at predicting NO3, as other modelling studies supported by in-situ measurements have shown.

**Response**: 1. We are grateful for this comment. The observations of radicals are important for assessing the reliability of box model. However, the relevant instruments are quite expensive and most of box model studies were carried out without those observations. To assess the reliability of our model results without radical measurements, we calculated the OH concentrations by using the ratio of two aromatics (ethylbenzene (E) and m,p-xylene (X)) that share common emission sources but with different reactivities with OH radicals as follows:

$$Ethylbenzene + OH \rightarrow products \quad k_{Ethylbenzene,OH} = 7.0 \times 10^{-12}$$

(R1)

$$m,p - Xylene + OH \rightarrow products \quad k_{m,p-Xylene,OH} = 1.89 \times 10^{-11}$$

(R2)

Therefore, mixing ratios of E and X at the sampling time can be expressed as follows:

$$[X]_t = [X]_0 \times e^{-[OH] \times k_{X,OH} \times t} \times f_{d,B}$$

(Eq.1)

where $[X]_0$ and $[X]_t$ are the mixing ratio of X at the initial time and after transport time t. $k_{X,OH}$ represents the temperature dependent reaction rate coefficient of m,p-xylene with OH, which was taken from the IUPAC database (http://iupac.pole-ether.fr/). $f_{d,B}$ represents the dilution factor of m,p-xylene in the atmosphere.

In this study, we assume that the rates of turbulent mixing and horizontal convection are similar for E and X. Therefore, during the transport time $\Delta t$, the dilution factor of E and X are the same. Then rearranging Eq.1 and extend this analysis to E and X will yield the following equation:

$$[OH]_{\frac{E}{X}} = \frac{1}{t \times (k_{E,OH} - k_{X,OH})} \times [ln\left(\frac{[E]}{[X]}\right)_0 - ln(\frac{[E]}{[X]})_t]$$

(Eq. 2)

where $[OH]_{E/X}$ is the estimated regional mixing ratio of OH based on ethylbenzene and m,p-xylene ratio.

The calculated average regional concentrations of OH ($8.39 \pm 5.11 \times 10^6$ molecules cm$^{-3}$) was in the same magnitude of the OBM-simulated result ($4.59 \pm 5.11 \times 10^6$ molecules cm$^{-3}$), suggesting that the OBM-simulated radical concentration is reliable.

**Relative description has been added in the revised manuscript (Page 8-9, Line 170-185):**

"To assess the reliability of model results without OH observation, we compared the OBM-simulated OH concentration with that calculated using the ratio of ethylbenzene (E) and m,p-xylene (X) that share common emission sources but with different reactivity with OH radicals (shown in Equation (5)~(8)):

$$Ethylbenzene + OH \rightarrow products \quad k_{Ethylbenzene,OH} = 7.0 \times 10^{-12} \quad (5)$$

$$m,p - Xylene + OH \rightarrow products \quad k_{m,p-Xylene,OH} = 1.89 \times 10^{-11} \quad (6)$$

$$[X]_t = [X]_0 \times e^{-[OH] \times k_{X,OH} \times t} \times f_{d,B} \quad (7)$$

$$[OH]_{\frac{E}{X}} = \frac{1}{t \times (k_{E,OH} - k_{X,OH})} \times [ln\left(\frac{[E]}{[X]}\right)_0 - ln(\frac{[E]}{[X]})_t] \quad (8)$$

where $[X]_0$ and $[X]_t$ are the mixing ratio of X at the initial time and after transport time t. $k_{X,OH}$ is the temperature dependent reaction rate coefficient of m,p-xylene with OH, which was taken from the IUPAC database (http://iupac.pole-ether.fr/), whereas $f_{d,B}$ is the dilution factor of m,p-xylene in the atmosphere. In this study, we assume that the rates of turbulent mixing and horizontal convection are similar for E and X. Therefore, during the transport time $\Delta t$, the dilution factor of E and X are the same. Therefore, rearranging Equation (7) and extending this analysis to E and X will yield Equation (8), where $[OH]_{E/X}$ is the estimated regional mixing ratio of OH by ethylbenzene and m,p-xylene ratio. The calculated average regional concentrations of OH ($8.39 \pm 5.11 \times 10^6$ molecules cm$^{-3}$) was in the same magnitude of the OBM-simulated result ($4.59 \pm 5.11 \times 10^6$ molecules cm$^{-3}$), suggesting that the OBM-simulated radical concentration is reliable."

2. We agree that the box model didn't performance well enough to predict nocturnal NO$_3$. Since this paper mainly focuses on daytime photochemical processes, we have removed the discussions about nocturnal chemistry.

4. I don't think the authors have made a compelling case that the chemistry in this area is driven by isoprene, if it is the intention of the paper to demonstrate this. Sure, isoprene is important and affects the formation of ozone and formaldehyde, but does it really dominate over all other VOCs, under these conditions? Before focusing on isoprene, the authors need to show evidence that other VOCs do not contribute as much to the oxidative capacity.

**Response**: We are grateful for the comment. We agree that isoprene is not the dominate VOCs compared with anthropogenic VOCs. According to observational data, the top 10 abundant VOC species during the observation period are: propane, toluene, ethylene, ethylbenzene, n-butane, ethane, i-butane, acetylene, xylene, i-pentane, and propene. Therefore, anthropogenic VOCs (such as alkanes and benzene series) are more abundant than isoprene. But the aim of this paper is to address the significance of isoprene chemistry in local photochemical processes. Hence, we tested the sensitivity of modelled O$_3$, HCHO, and OH concentrations without EXT (ethylbenzene, xylene, and toluene), alkenes (acetylene, ethylene, and propene), and isoprene. The sensitivity to the presence of alkanes was not conducted since they are relatively inert. Although the averaged isoprene concentration was only 0.37 ± 0.36 ppbv, cutting isoprene input can lead to obvious drop in simulated O$_3$, HCHO, and OH concentration, which was comparable to that of cutting EXT and alkenes, indicating the significant role of isoprene in local photochemical processes.

**Relevant description has been added in Page 20, Line 403-410:**

"To compare the importance of isoprene and other abundant VOCs in local chemistry at DSL site, sensitivity analysis was conducted for the modelled O$_3$, HCHO, and OH concentrations without the input of active VOCs (toluene, ethylene, ethylbenzene, ethane, acetylene, xylene, propene, and isoprene). Results suggested that, although the average isoprene concentration was only 0.37 ± 0.36 ppbv, cutting isoprene input can lead to obvious drop in simulated O$_3$, HCHO, and OH, which was comparable to that of cutting EXT and alkenes, indicating the significant role of isoprene in local photochemical processes (Figure S6)."

[Figure]

**Figure S6.** Sensitivity analysis of OBM modelled O₃, HCHO, and OH concentrations without alkenes (including ethylene, propene, and acetylene), isoprene, and EXT (ethylbenzene, xylene, and toluene) input.

**Point-by-point response to comments by Reviewer#2**

The revised manuscript has addressed most of the comments raised by peer reviewers but a few technical questions remain, as follows:

We thank the reviewer for the detailed and constructive review comments. Below is our point-by-point response to each comment, marked in blue. Changes made to the main text are also marked in blue in the revised manuscript file.

**Comment 1:** Isoprene measurement should be described or referenced since this species is a key VOC in the manuscript. In addition, other biogenic VOC emissions such as terpenes can be important in terms of contributing to ozone and formaldehyde formation but they are not discussed in the manuscript.

**Response**: Thank you for raising this issue. We agree that BVOC species like isoprene and terpenes are important precursors of ozone and formaldehyde. Among them, isoprene represents the dominant BVOC specie with highest emissions and is therefore the most commonly used indicator of biogenic emissions in terms of VOCs measurement. Description of the measurement of isoprene has been inserted in the revised manuscript. We also added discussions with respect to other BVOCs species in the revised manuscript.

**Revised manuscript (Page 6, Line 116-120):**
A total of 55 VOC species, including 28 alkanes, 10 alkenes (including isoprene), 16 aromatics and acetylene were continuously analyzed at our sampling site by two online gas chromatographs with flame ionization detector (GC-FID) systems (GC-866 airmoVOC C2-C6 #58850712 and airmoVOC C6-C12 #283607112, Agilent., USA) with a time resolution of 1 hour during the study period."
**Revised manuscript (Page 26, Line 531-534):**
"Furthermore, other biogenic VOCs (BVOCs, such as terpene and sesquiterpene) can also affect local chemistry via photochemical processes, but those BVOCs were not able to be synchronously observed. Therefore, future studies should take into account those BVOCs."

**Comment 2:** Lines 80-82: Not sure what "trace gases" were collected to understand the impact of isoprene since no key products of isoprene photochemical reaction were observed or measured.

**Response**: To avoid misunderstanding, we have revised this sentence as following:

In this study, we conducted a comprehensive set of in-situ observations of isoprene, meteorological parameters, and atmospheric pollutants (including $O_3$, NOx, CO, VOCs, and HCHO) to understand the important impact of isoprene chemistry on atmospheric photochemical processes in suburban YRD region.

**Comment 3:** Line 153: Figures S1 and S2 do not correspond to the correct graphs, respectively.

**Response**: Thanks for raising this issue. We have corrected this problem in supporting information and in the revised manuscript. Please refer to Page 8, Line 156-157: " The comparison of simulated and observed $O_3$ and HCHO concentrations is shown in Figure S1 and Figure S2."

**Comment 4:** Lines 89-90: It states that DSL has high vegetation coverage, implying more biogenic emissions but that is not the case from Figure 5 of Lin et al (2020).

**Response**: As suggested by Figure 5 of Lin et al. (2020), the RIR(NHC) is relatively higher than RIR(AHC) at DSL site. Therefore, our previous description "It is located in the west of Shanghai and is close to the Dianshan Lake Scenic area, which has high vegetation coverage" is inappropriate. However, it should be noted that RIR only stands for sensitivity but not the amount of emissions. We have corrected this sentence as "It is located in the west of Shanghai and is close to the Dianshan Lake Scenic area, which has relatively higher vegetation coverage than the urban areas."

**Comment 5:** Figure S2: The observed and simulated formaldehyde concentrations are less correlated than ozone with the reasons being uncertainties in dry deposition, vertical transport, and so on. Did the authors learn something new or different from these reasons? How did the model treat primary HCHO emissions from secondary? The overall HCHO discussion is useful but diminished due to the inconsistency.

**Response**: Thanks for this constructive comment. Apart from dry deposition and vertical transportation, primary HCHO emissions can also bring uncertainties in the simulation. Due to

the lack of HCHO sources for areas around DSL, primary HCHO emissions were not included in our model.

**Relevant descriptions have been added in Page 8, Line167-170:**

"In addition, primary HCHO sources can contributed up to 76% of total HCHO concentration in urban areas (Li et al., 2010). However, due to the lack of primary HCHO sources for areas around DSL, primary HCHO emissions were not included in our model. "

**Comment 6:** It is not convincing that NO peak occurred in early morning, indicating traffic pattern, while CO was flat. Is this instrument sensitivity issue?

**Response**: Thanks for the reviewer's careful comment. The sensitivity of CO analyzer was set to 0.1 mg/m$^3$, which was about 80 ppbv during our observation. This coarse resolution caused the flat CO variations during our observation. The other potential reason is that CO is mainly emitted by gasoline vehicles while NOx are mainly from heavy-duty vehicle exhausts. Since DSL site is far away from urban area, there are rare passenger cars in early morning whereas heavy-duty trucks sometimes pass by, causing NO peaks in early morning.

**For better explanation, we have added relevant descriptions in the revised manuscript (Page 10, Line 215-221):**

"It should also be noted that, flat CO pattern was found during morning when NO$x$ peaks were observed. This inconformity can be attributed to the coarse resolution of CO analyzer (about 80 ppbv) and CO emission source  (mainly gasoline vehicles) while NO$x$ are mainly emitted by heavy-duty vehicle exhausts. Therefore, since DSL site is far from urban area, it is unlikely to have gasoline vehicles in early morning. On the contrary, there are sometimes heavy-duty trucks passing by, causing peaks of NO in early morning."

Besides the above-mentioned revision, we also thoroughly polished the language to improve the quality of the paper.

---

## Author Response (AR3)

**Point-by-point response to comments by editor**

We thank the editor for the detailed and constructive review comments. Below is our point-by-point response to each comment, marked in blue.

**Comments:**

(1) For this section on page 9 in the revised MS

$Ethylbenzene + OH \rightarrow products\ kEthylbenzene,OH = 7.0 \times 10{-}12$ (R1)

$m, p - Xylene + OH \rightarrow products\ km,p{-}Xylene,OH = 1.89 \times 10{-}11$ (R2)

Please insert the units of the rate coefficients.

**Response**: Thanks for raising this issue. We have inserted the units of these rate coefficients ($cm^3$ molecule$^{-1}$ s$^{-1}$) in the revised manuscript. Please refer to Page 9:

$$Ethylbenzene + OH \rightarrow products$$
$$k_{Ethylbenzene,OH} = 7.0 \times 10^{-12} (cm^3\ molecule^{-1}\ s^{-1}) \quad (5)$$

$$m, p - Xylene + OH \rightarrow products$$
$$k_{m,p-Xylene,OH} = 1.89 \times 10^{-11} (cm^3\ molecule^{-1}\ s^{-1}) \quad (6)$$

(2) Also, in Table 2, caption, spelling of Heterogenous should be "Heterogeneous".

**Response**: Thanks for raising this issue. We have revised this spelling error. Please refer to Page 7, Line 143: "Table 2. Heterogeneous reactions and associated rate constants used in the OBM model."